# *NORFA*, long intergenic noncoding RNA, maintains sow fertility by inhibiting granulosa cell death

Xing Du[1], Lu Liu[1], Qiqi Li[1], Lifan Zhang[1], Zengxiang Pan[1] & Qifa Li [1✉]

Long intergenic non-coding RNAs (lincRNAs) have been proved to be involved in regulating female reproduction. However, to what extent lincRNAs are involved in ovarian functions and fertility is incompletely understood. Here we show that a lincRNA, *NORFA* is involved in granulosa cell apoptosis, follicular atresia and sow fertility. We found that *NORFA* was down-regulated during follicular atresia, and inhibited granulosa cell apoptosis. NORFA directly interacted with miR-126 and thereby preventing it from binding to *TGFBR2* 3'-UTR. miR-126 enhanced granulosa cell apoptosis by attenuating *NORFA*-induced TGF-β signaling pathway. Importantly, a breed-specific 19-bp duplication was detected in *NORFA* promoter, which proved association with sow fertility through enhancing transcription activity of *NORFA* by recruiting transcription factor NFIX. In summary, our findings identified a candidate lincRNA for sow prolificacy, and provided insights into the mechanism of follicular atresia and female fertility.

[1] College of Animal Science and Technology, Nanjing Agricultural University, 210095 Nanjing, China. ✉email: liqifa@njau.edu.cn

Long non-coding RNAs (lncRNAs) are defined as a class of RNAs whose transcripts are >200 nucleotides (nt) without open reading frame or protein-coding potential. After microRNAs (miRNAs), lncRNAs are rapidly emerging as important ncRNA, which have been proved to be distinctly involved in a wide range of biological and cellular processes in development and disease[1–3]. Compared with protein-coding RNAs, lncRNAs are often modest evolutionary conservation, unstable, relatively lower abundance in vivo, and tighter tissue-specific expression, which result in difficulties in lncRNAs research[4,5]. In recent years, thousands of lncRNAs have been identified to be actively transcribed from human genome and other organisms by high-throughput RNA sequencing (RNA-seq) technology and bioinformatics[6,7], but only a small number of lncRNAs are screened in domestic animals, and well characterized in humans and rodents[8,9]. Therefore, the biological functions, mode of action, and regulation of most lncRNAs are still largely unknown.

It is known that mature miRNAs are mainly located in the cytoplasm, which function through degrading target mRNAs or inhibiting target mRNA translation at the posttranscriptional level[10]. In contrast to miRNAs, lncRNAs can be located either in the cytoplasm or nucleus, with 17% of lncRNAs showing relative enrichment in the nucleus and 26% relatively enriched in the cytoplasm[11,12]. Notably, different subcellular localization (nuclear or cytoplasm) of lncRNAs may participate in different mechanisms[13,14], so the mechanism of lncRNAs action is more complex than that of miRNAs. For example, lncRNAs retained in the nucleus carry out their biological functions by interacting with chromatin-modifying complexes and transcriptional regulatory proteins[15,16], directly controlling gene transcription in cis or trans[17] and scaffolding of nuclear complexes[18]. Cytoplasmic lncRNAs are known to affect the activity or stability of mRNA through pairing with other RNA molecules such as mRNAs[19] and miRNAs[20], scaffolding of cytoplasmic complexes[21], and decoying proteins and miRNAs away from its potential targets[22].

miRNAs have been shown to target ovarian function-related genes and control various ovarian function such as steroidogenesis[23], folliculogenesis[24], ovulation[25], and luteogenesis[26]. However, little is known of the role of lncRNAs in ovarian function[27]. In this study, we prove that NORFA (non-coding RNA involved in the follicular atresia), a long intergenic non-coding RNA (lincRNA), controls porcine granulosa cell apoptosis and follicular atresia by acting as a competing endogenous RNA (ceRNA) and inhibits endogenous miR-126. We also demonstrate that NORFA/miR-126 axis plays an important role in regulating granulosa cells apoptosis through targeting the transforming growth factor-β (TGF-β) signaling pathway. Furthermore, we identified a breed-specific 19-bp duplication in NORFA promoter, which could regulate NORFA transcription by altering the recruitment of NFIX to the promoter of NORFA.

## Results

### Identification and characterization of NORFA.
Our recent study performed deep transcriptome sequencing in SMAD4-silencing porcine granulosa cells[28], and a transcript that was highly expressed in granulosa cells was noted. We then isolated the full-length sequence of the transcript by using 5′ and 3′ rapid amplification of cDNA ends (RACE) (Supplementary Fig. 1) and identified the full length of this transcript to be 739 nt (Fig. 1a). Moreover, the transcript was found to be located between TRAF2 gene and EDF1 gene at pig chromosome 1, which consisted of 2 exons (Fig. 1b). The homologous sequence of this transcript was not detected in the genome of other mammals and the RNA structure is low conserved (Supplementary Fig. 2), suggesting that

it is a pig-specific transcript. Both PhyloCSF and CPAT analysis showed that the transcript is a lincRNA, with the low coding potential, similar to other well-characterized lncRNAs such as HOTAIR and H19 (Fig. 1c, d).

The transcript had a low expression level in the duodenum, muscle adipose, and colon, while high expression was found in the lung, kidney, and ovary, more especially in the ovary (Fig. 1e). In porcine ovary, fluorescence in situ hybridization (FISH) showed that the transcript was mainly expressed in granulosa cells of follicles at different stages (preantral follicles, antral follicles, and mature follicles), while its expression was dramatically reduced in granulosa cells of atresia follicles (Fig. 1f). Moreover, quantitative reverse transcriptase polymerase chain reaction (qRT-PCR) revealed that the transcript was downregulated during follicular atresia of porcine ovary (Fig. 1g, h), indicating that the transcript might play an important role in follicular atresia of porcine ovary. Therefore, the transcript was termed as NORFA.

### NORFA is involved in granulosa cell apoptosis.
To further investigate the role of NORFA in follicular atresia, we synthesized NORFA expression vector and NORFA-specific small interfering RNA (siRNA) to overexpress and knockdown endogenous NORFA in granulosa cells cultured in vitro, respectively. We found that overexpression of NORFA enhanced NORFA level in granulosa cells (Fig. 2a). Besides, we also noticed that the expression level of pro-apoptotic gene BAX was decreased (Fig. 2b), and anti-apoptotic gene BCL-2 mRNA level (Fig. 2c) and BCL-2/BAX ratio (Fig. 2d) were upregulated after NORFA overexpression. In addition, NORFA overexpression decreased cell apoptosis rate (10.97 ± 0.58% vs 6.40 ± 0.57%) (Fig. 2e), indicating that NORFA is an anti-apoptotic factor in granulosa cells. By contrast, knockdown of NORFA attenuated NORFA levels (Fig. 2f) and BAX mRNA levels (Fig. 2g) but decreased BCL-2 mRNA levels (Fig. 2h) and BCL-2/BAX ratio (Fig. 2i). Furthermore, knockdown of NORFA increased cell apoptosis rate (9.03 ± 0.55% vs 13.86 ± 0.23%) (Fig. 2j). All our data suggest that NORFA is essential for inhibiting granulosa cell apoptosis and is involved in follicular atresia of pigs.

### NORFA acts as a ceRNA of its nearby gene miR-126.
To explore the functional mechanism of NORFA in porcine granulosa cells, we determined the effects of NORFA on the expression levels of nearby genes including 16 coding genes and 1 miRNA gene (miR-126) (Fig. 3a). The expression levels of four coding genes (EGFL7, PHPT1, TMEM141, and LCN10) were increased in granulosa cells after NORFA overexpression (Fig. 3b) but decreased after NORFA silencing (Fig. 3c). Interestingly, miR-126, an intronic miRNA transcript from EGFL7[29], was dramatically downregulated by NORFA overexpression (Fig. 3d) and upregulated by NORFA silencing in granulosa cells (Fig. 3e). These data suggested NORFA regulates the expression of its nearby gene including coding genes and the miRNA gene.

We next analyzed the cellular localization of NORFA in granulosa cells. High levels of cytoplasmic markers (glyceraldehyde 3-phosphate dehydrogenase (GAPDH) and RPLP0) were detected in cytoplasm extraction, whereas the isolated nuclear fraction displayed a high level of nuclear marker (U6) (Fig. 3f). It should be noted that >75% of NORFA was located in the cytoplasm (Fig. 3f), indicating that NORFA was a cytoplasmic lncRNA, which was consistent with FISH assay (Fig. 3g; Supplementary Fig. 3a, b). Interestingly, an miRNA response element (MRE) of miR-126 was predicted at 556–578 nt of the NORFA transcript (Fig. 3h). To confirm whether miR-126 directly binds to NORFA, dual-luciferase reporters containing

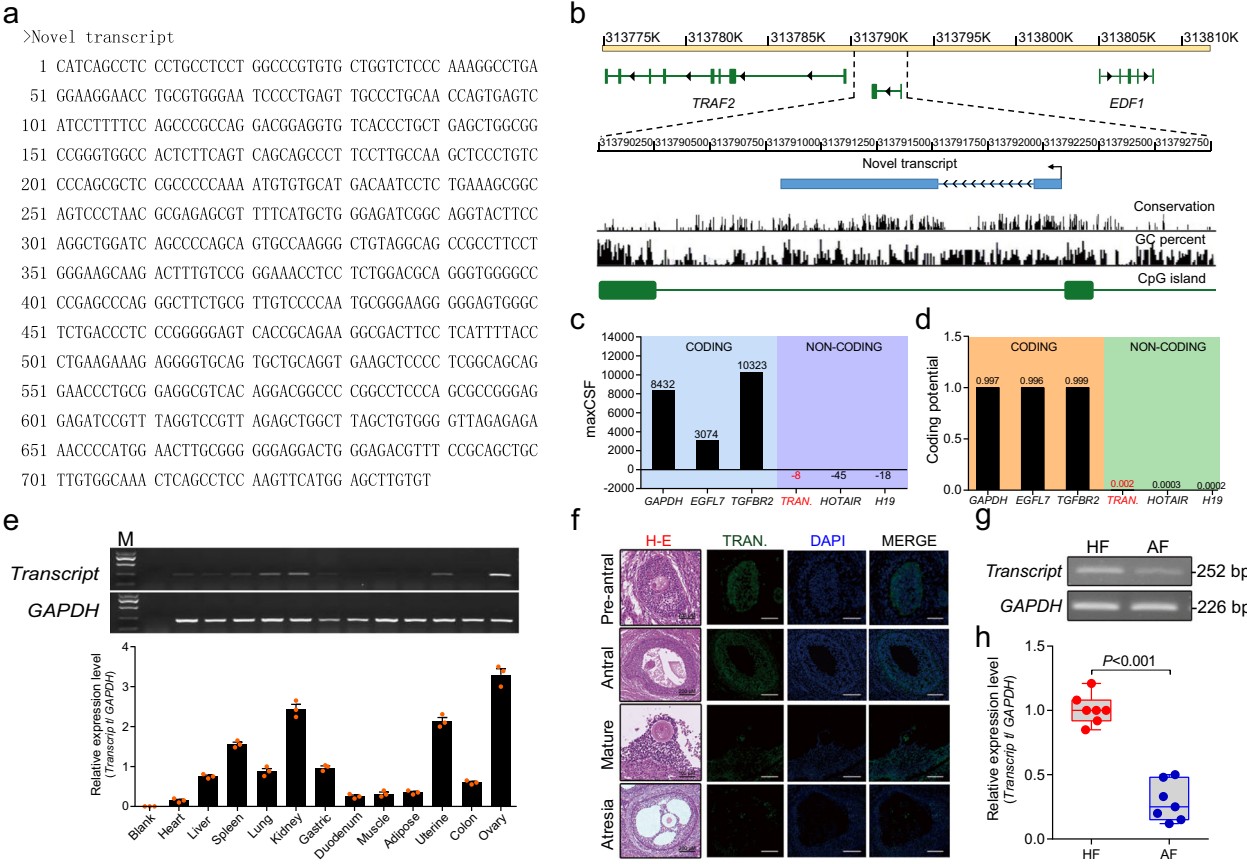

**Fig. 1 Identification and characterization of a transcript in pigs. a** The full-length RNA sequence of the transcript. The sequence of the transcript was isolated from porcine granulosa cells by using 5′- and 3′-RACE (see also Supplementary Fig. 1). **b** Schematic representation of the transcript with associated UCSC Genome Browser tracks depicting mammalian conservation, GC percent, and CpG Islands. **c** Maximum codon substitution frequency scores of the transcript as well as other known coding RNAs (*GAPDH*, *EGFL7*, and *TGFBR2*) and non-coding RNAs (*HOTAIR* and *H19*) were measured by PhyloCSF; TRAN. indicates the transcript. **d** Coding potential of the transcript and other RNAs were calculated by CPAT. **e** Tissue expression patterns of the transcript in female pigs (n = 3). **f** H–E staining (array 1) and FISH staining (array 2–4) of the follicles in porcine ovary. Nuclear were stained by DAPI dye (blue). The transcript was shown as green (Scale bars, 200 μm for lines 1, 2, and 4 and 100 μm for line 3). **g, h** The expression levels of the transcript in granulosa cells from healthy follicles (HF) and atretic follicles (AF) were measured by qRT-PCR (n = 7). Data in **e** are shown as mean ± S.E.M. with three independent experiments.

the MRE (wild-type (wt)) and mutant plasmid were generated (Fig. 3i) and co-transfected with miR-126 mimics into HEK293T-cells. Luciferase activity of wt construct was dramatically decreased, but no change was observed in mutant-type construct (Fig. 3j), indicating that direct interaction exist between miR-126 and *NORFA* MRE motif. To further confirm the physical association between *NORFA* and miR-126, RNA pull-down assays were performed and confirmed that *NORFA* physically interacted with miR-126 in porcine granulosa cells (Fig. 3k; Supplementary Fig. 3c–f). In addition, we also assessed whether the expression level of *NORFA* was regulated by miR-126 in granulosa cells. However, there was no obvious changes in *NORFA* level after overexpression or knockdown of miR-126 in granulosa cells (Fig. 3l, m). All the data are in line with the notion that *NORFA* acts as a ceRNA and sponges endogenous miR-126 in granulosa cells.

**miR-126 is relevant to *NORFA* expression in follicles.** Owing to the unavailability of information on the characterization of the gene encoding miR-126 in pig, we first isolated and characterized the porcine gene encoding miR-126 (ssc-miR-126). The length of ssc-miR-126 gene is 73 bp, which shares high nucleotide identities with other mammals, such as humans, and other vertebrates, such as chicken and zebrafish (Supplementary Fig. 4). Furthermore,

the mature sequence of ssc-miR-126 (ucguaccgugaguaauaaugcg) and the seed region (cguaccg) are completely consistent with that in other vertebrates, respectively (Fig. 4a). Vertebrate miR-126 containing ssc-miR-126 is an intronic miRNA, which is embedded within the introns of gene encoding protein EGFL7 (Fig. 4b).

To understand the potential function of miR-126, Kyoto Encyclopedia of Genes and Genomes (KEGG) analysis was performed to check out the pathways from miR-126 targets. The targets of miR-126 are enriched in the apoptosis, pathways in cancer, and phosphoinositide-3 kinase-AKT signaling pathway (Fig. 4c). Our previous study showed that miR-126 was upregulated during follicular atresia in porcine ovary identified by miRNA microarray assay[30] (Fig. 4d), which is consistent with the present observation from qRT-PCR (Fig. 4e). The correlation analysis showed that the miR-126 levels in follicles had a negative correlation with *NORFA* levels (Fig. 4f). All the data clearly indicated that intronic miR-126 gene is highly evolutionary conserved in vertebrates and correlates with *NORFA* levels in follicles of porcine ovary.

**NORFA inhibits the pro-apoptotic activity of miR-126.** Next, we analyzed the role of miR-126 in porcine granulosa cells. miR-126 was increased after treatment with miR-126 mimics (Supplementary Fig. 5a), then the rate of cell apoptosis was

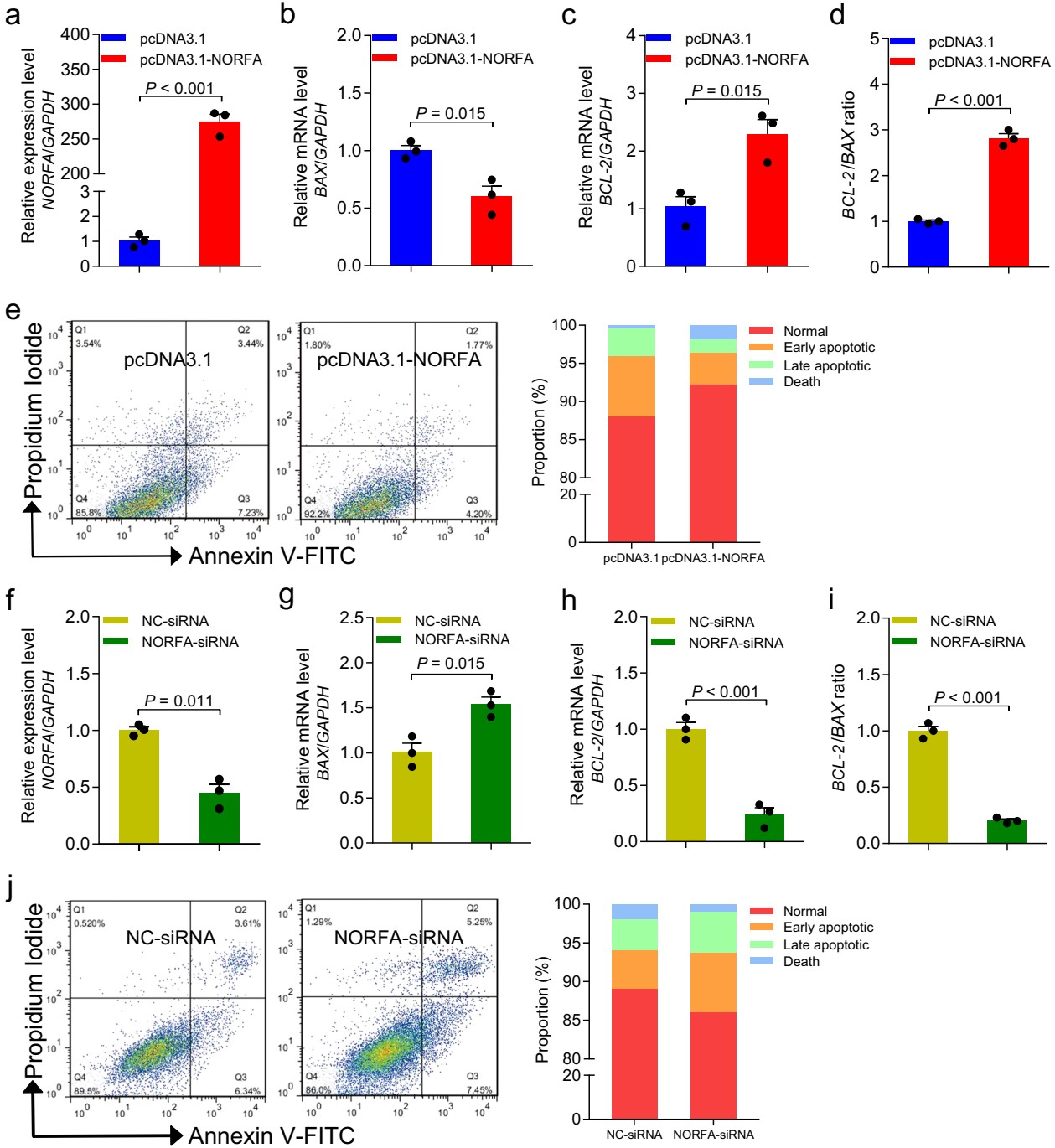

**Fig. 2 NORFA inhibits porcine granulosa cell apoptosis. a–e** Porcine after transfection with pcDNA3.1-*NORFA* for 24 h, and the expression levels of *NORFA* (**a**), *BAX* (**b**), and *BCL-2* (**c**) were detected by qRT-PCR. *BCL-2/BAX* ratio (**d**) was calculated, and the apoptosis rate (**e**) was determined by FACS. **f–j** *NORFA* (**f**), *BAX* (**g**), and *BCL-2* (**h**) expression after *NORFA* silencing were measured by qRT-PCR, *BCL-2/BAX* ratio (**i**) was calculated, and granulosa cell apoptosis rate (**j**) was determined by FACS. Data in **a–d** and **f–i** are represented as mean ± S.E.M. with three independent experiments. *P* values were calculated by a two-tailed Student's *t* test.

increased (9.08 ± 0.32% vs 16.75 ± 2.01%) (Fig. 5a) and *BCL-2/BAX* ratio was decreased (Fig. 5b). We also knocked down miR-126 by transfecting a miR-126 inhibitor in vitro (Supplementary Fig. 5b). The data showed that the apoptosis rate of granulosa cells (15.59 ± 3.06% vs 8.16 ± 0.11%) was dramatically decreased (Fig. 5c) and *BCL-2/BAX* ratio was dramatically increased simultaneously (Fig. 5d) after miR-126 silencing. These data suggested that miR-126 is a pro-apoptotic epigenetic regulator in granulosa cells.

To further investigate whether miR-126 mediated the function of *NORFA*, miR-126 and *NORFA* were overexpressed or knocked down by co-transfecting a miR-126 mimic and pcDNA3.1-*NORFA* or miR-126 inhibitor and *NORFA*-specific siRNA. The results showed that miR-126 overexpression prevents *NORFA*-reduced cell apoptosis (6.39 ± 0.58% vs 12.19 ± 0.66%) and *NORFA*-induced *BCL-2/BAX* ratio (Fig. 5e, f). In contrast, inhibition of miR-126 rescued cell apoptosis (13.82 ± 0.61% vs 8.74 ± 0.75%) caused by *NORFA*-specific siRNA (Fig. 5g) and

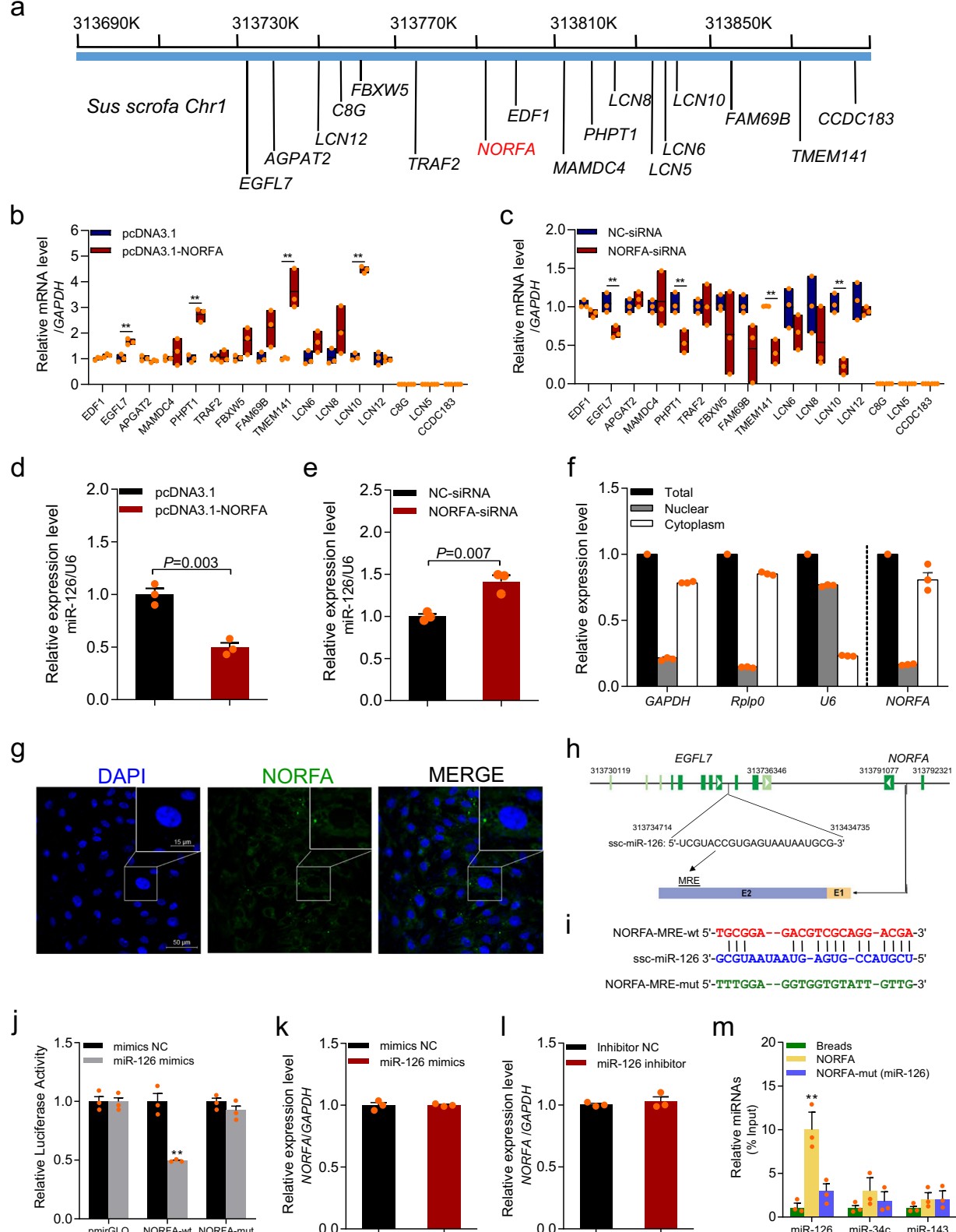

*NORFA*-siRNA reduces *BCL-2*/*BAX* ratio (Fig. 5h). These data reveal that *NORFA* inhibits cell apoptosis through sponging miR-126 in porcine granulosa cells.

**TGFBR2 (TGF-β receptor 2) is a direct target of miR-126 in granulosa cells.** To evaluate the molecular mechanism by which miR-126 affects granulosa cell apoptosis and follicular atresia,

four different algorithms were performed to predict the targets of miR-126. As a result, four common targets (*PIK3R2*, *PEX5*, *TGFBR2*, and *TNFRSF10B*) were identified (Fig. 6a). Of them, *TGFBR2* is an important component of TGF-β signaling pathway that is involved in porcine granulosa cell apoptosis and follicular atresia[31]. MRE motif of miR-126 was located at the residues 484–506 nt in the porcine *TGFBR2* 3′-untranslated region (UTR).

**Fig. 3 NORFA acts as a ceRNA and sponges miR-126 in porcine granulosa cells. a** Schematic showing the locations of NORFA (red) and its nearby coding genes on porcine chromosome 1. **b, c** The expression levels of *NORFA* nearby coding genes in porcine granulosa cells were detected by qRT-PCR after transfection with pcDNA3.1-*NORFA* (**b**) or *NORFA*-siRNA (**c**). **d, e** The expression levels of miR-126 in porcine granulosa cells were measured after pcDNA3.1-*NORFA* (**d**) or *NORFA*-siRNA (**e**) treatment. Level of miR-126 was detected by stem-loop qRT-PCR. **f** Subcellular localization of *NORFA* in porcine granulosa cells. Expression levels of *NORFA* and marker gene (*GAPDH* and *RPLP0* for cytoplasm and *U6* for nuclear) in isolated nuclear and cytoplasm fraction from porcine granulosa cells were detected by qRT-PCR. **g** Subcellular localization of *NORFA* in porcine granulosa cells was detected by FISH (Scale bars, 50 μm). Nucleus was dyed with DAPI (blue) and NORFA was dyed with NORFA-specific probe (green). **h** Schematic showing an miRNA response element (MRE) of miR-126 in exon 2 (E2) of porcine NORFA. **i** Schematic showing the interactions of miR-126 with wild-type NORFA (red) and the mutant version (green). **j** Porcine granulosa cells were co-transfected with dual-luciferase reporter vector containing the wild-type *NORFA* or the mutant version and miR-126 mimics, and luciferase activity was detected. **k, l** qRT-PCR analysis showing the expression levels of NORFA in porcine granulosa cells when miR-126 was overexpressed (**k**) or silenced (**l**). **m** Porcine granulosa cells lysates were incubated with biotin-labeled wild-type or mutated NORFA. After pull-down, miRNAs were quantified by qRT-PCR. miR-34c and miR-143 were used as negative controls. Experiments were conducted in triplicate. Data in **b**–**e** and **j**–**m** are shown as mean ± S.E.M. with three independent experiments. *P* values were calculated by a two-tailed Student's *t* test. \*\**P* < 0.01.

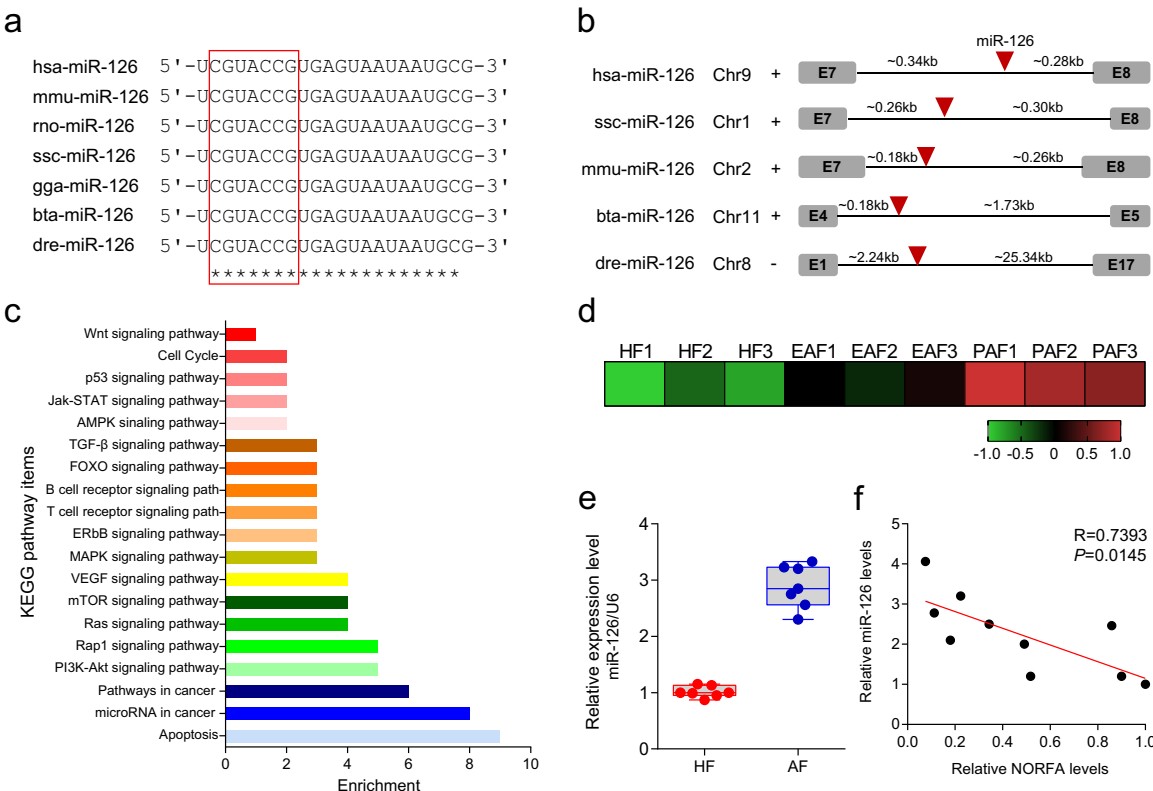

**Fig. 4 Characterization of porcine miR-126. a** miR-126 mature sequences are highly conserved among vertebrates. Asterisks indicate complementarity. Red box indicate seed sequence of miR-126. ssc *Sus scrofa*, hsa *Homo sapiens*, mmu *Mus musculus*, rno *Rattus norvegicus*, gga *Gallus gallus*, bta *Bos taurus*, dre *Danio rerio*. **b** Genomic organization of miR-126 in pig and other species. The approximate distances between miRNAs and nearby protein-coding genes are given. Chromosome strands are indicated by + or −. **c** KEGG pathway analysis with miR-126 targets. **d** Heatmap of miR-126 levels during porcine ovarian follicular atresia. The data were identified by miRNA microarray assay[30]. HF healthy follicles, EAF early atresia follicular, PAF progressively atretic follicles. **e** miR-126 levels in granulosa cells from healthy follicles (HF) and atretic follicles (AF) (*n* = 7), analyzed by qRT-PCR. **f** The relationship between *NORFA* and miR-126 expression levels in porcine ovarian follicles was detected with Pearson analysis.

Furthermore, miR-126 seed sequence was completely paired with the 3′-UTR of vertebrate *TGFBR2* (Fig. 6b).

To investigate whether *TGFBR2* is a target of miR-126, luciferase reporters containing the wild MRE motif or mutation were constructed (Fig. 6c) and co-transfected with miR-126 mimics into HEK293T cells. Luciferase assay showed that miR-126 decreased luciferase activity of the reporter containing the wt MRE motif (Fig. 6d) and with no change of reporter with the mutant MRE motif (Fig. 6e), indicating that *TGFBR2* is a direct target of miR-126. We further examined the effects of miR-126 on the expression of TGFBR2 in granulosa cells cultured in vitro. It was found that ectopic expression of miR-126 attenuated the

TGFBR2 expression in both the mRNA and protein levels (Fig. 6f, h), while inhibition of miR-126 enhanced the TGFBR2 expression (Fig. 6g, i). These results implicate that *TGFBR2* is a functional target of miR-126 in porcine granulosa cells.

**miR-126 regulates cell death by targeting TGFBR2.** To test whether TGFBR2 is involved in follicular atresia of porcine ovary, we measured TGFBR2 levels in follicles during follicular atresia. Both mRNA and protein levels of TGFBR2 were downregulated during follicular atresia (Fig. 7a, b). Correlation analysis confirmed that *TGFBR2* mRNA levels had a negative correlation with

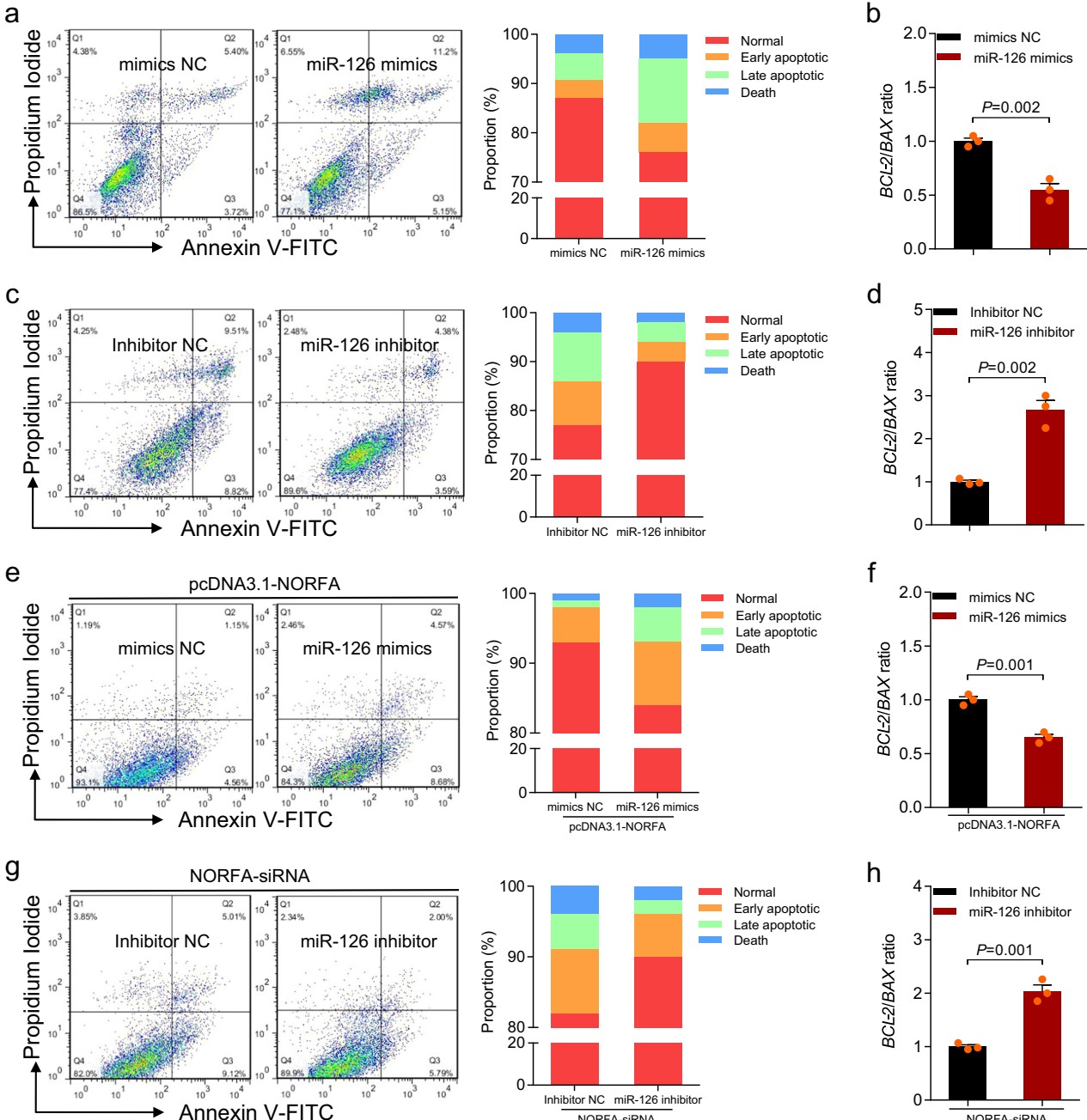

**Fig. 5 miR-126 regulates porcine granulosa cell apoptosis and mediated the functions of *NORFA*. a, b** Granulosa cells were transfected with miR-126 mimics, apoptosis rate was determined by FACS (**a**), and *BCL-2/BAX* ratio was calculated (**b**). **c, d** Granulosa cells were transfected with miR-126 inhibitor, apoptosis rate was determined by FACS (**c**), and *BCL-2/BAX* ratio was calculated (**d**). **e, f** Granulosa cells were co-transfected with pcDNA3.1-*NORFA* and miR-126 mimics, cell apoptosis was determined by FACS (**e**), and *BCL-2/BAX* ratio was calculated (**f**). **g, h** Granulosa cells were co-transfected with *NORFA*-siRNA and miR-126 inhibitor, apoptosis rate was determined by FACS (**g**), and *BCL-2/BAX* ratio was calculated (**h**). Data are represented as mean ± S.E.M. with three independent experiments. *P* value was calculated by a two-tailed Student's *t* test.

miR-126 levels in follicles (Fig. 7c), indicating that *TGFBR2* might have an opposite effect on follicular atresia compared to the function of miR-126.

Next, we analyzed the role of TGFBR2 in porcine granulosa cell apoptosis. The synthesized expression vector pcDNA3.1-TGFBR2 increased the TGFBR2 protein level (Fig. 7d) and reduced cell apoptosis rate (9.31 ± 0.47% vs 5.01 ± 0.33%; Fig. 7e). By contrast, *TGFBR2*-specific siRNA dramatically decreased the TGFBR2 protein level (Fig. 7f) and enhanced cell apoptosis (6.60 ± 0.32%

vs 10.25 ± 1.03%) (Fig. 7g), suggesting that TGFBR2 is an anti-apoptotic factor in porcine granulosa cells.

To assess whether TGFBR2 mediated miR-126-induced granulosa cell apoptosis, miR-126 mimics or inhibitor and pcDNA3.1-TGFBR2 or *TGFBR2*-specific siRNA were co-transfected into cultured granulosa cells in vitro. We found that cell apoptosis (9.94 ± 1.08% vs 6.54 ± 0.41%) induced by miR-126 was recovered when TGFBR2 was overexpressed (Fig. 7h). In contrast, inhibition of TGFBR2 weakened the anti-apoptotic effect of miR-126

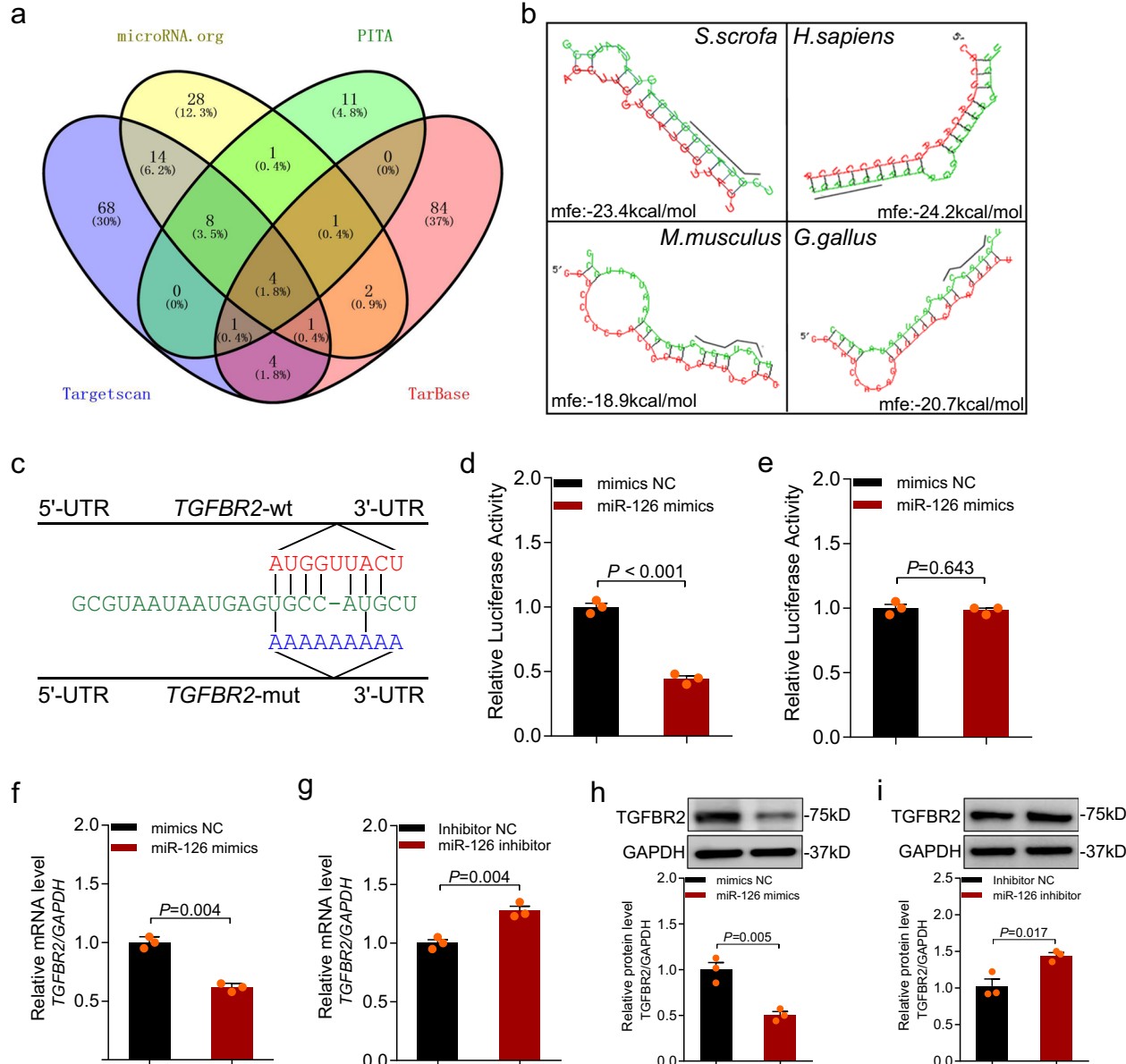

**Fig. 6 TGFBR2 is a direct target of miR-126 in porcine granulosa cells. a** Potential target genes of miR-126 were predicted by four different prediction algorithms (microRNA.org, PITA, Targetscan, and TarBase). **b** miRNA response elements (MREs) of miR-126 within *TGFBR2* 3′-UTR of four species (human, pig, mouse, and chicken) were predicted by RNAhybrid. mfe minimum free energy. **c** miR-126 MRE motif (red letters) and mutated type (blue letters) in the porcine *TGFBR2* 3′-UTR. **d, e** Porcine granulosa cells were co-transfected miR-126 mimics with dual-luciferase reporter vector containing the wild-type *TGFBR2* 3′-UTR (**d**) or the mutant version (**e**), and luciferase activity was detected. **f, g** *TGFBR2* mRNA levels in granulosa cells after transfection with miR-126 mimics (**f**) or inhibitor (**g**) were measured by qRT-PCR. Western blotting was used to determine the change of TGFBR2 protein levels in granulosa cells after treatment with miR-126 mimics (**h**) or inhibitor (**i**). Data in **d**–**i** are shown as mean ± S.E.M. from three independent experiments. *P* values were calculated by a two-sided Student's *t* test.

inhibitor (6.63 ± 0.30% vs 11.15 ± 0.13%; Fig. 7i). Together, these data provide strong evidence that miR-126 regulates granulosa cell apoptosis and follicular atresia by targeting *TGFBR2*.

**NORFA activates TGF-β pathway by miR-126–TGFBR2 axis.** We next detected whether *NORFA* acted as an upstream molecule of miR-126 and affected the TGFBR2 expression. Correlation analysis revealed that *NORFA* level in follicles was positively correlation with the *TGFBR2* mRNA level (Fig. 8a). In granulosa cells, we found that *NORFA* positively regulated the TGFBR2 expression at both the transcriptional level (Fig. 8b, c) and translational level (Fig. 8d, e), indicating that *NORFA* is a regulator of TGFBR2. To further determine whether miR-126

mediated this process, miR-126 was transfected into granulosa cells with pcDNA3.1-NORFA. Western blot showed that miR-126 inhibited *NORFA*-induced TGFBR2 expression (Fig. 8f). Besides, miR-126 inhibitor rescued *NORFA*-specific siRNA-reduced TGFBR2 expression (Fig. 8g). These data demonstrated that miR-126 mediated *NORFA* induction of TGFBR2 process in porcine granulosa cells.

To investigate whether *NORFA*–miR-126 axis interacts with TGF-β signaling pathway, we examined the total protein and phosphorylated levels of SMAD3, a core component of TGF-β signaling pathway and a downstream of TGFBR2. We found that *NORFA* overexpression increased p-SMAD3 (phospho-SMAD3) levels (Fig. 8h), whereas the p-SMAD3 was dramatically

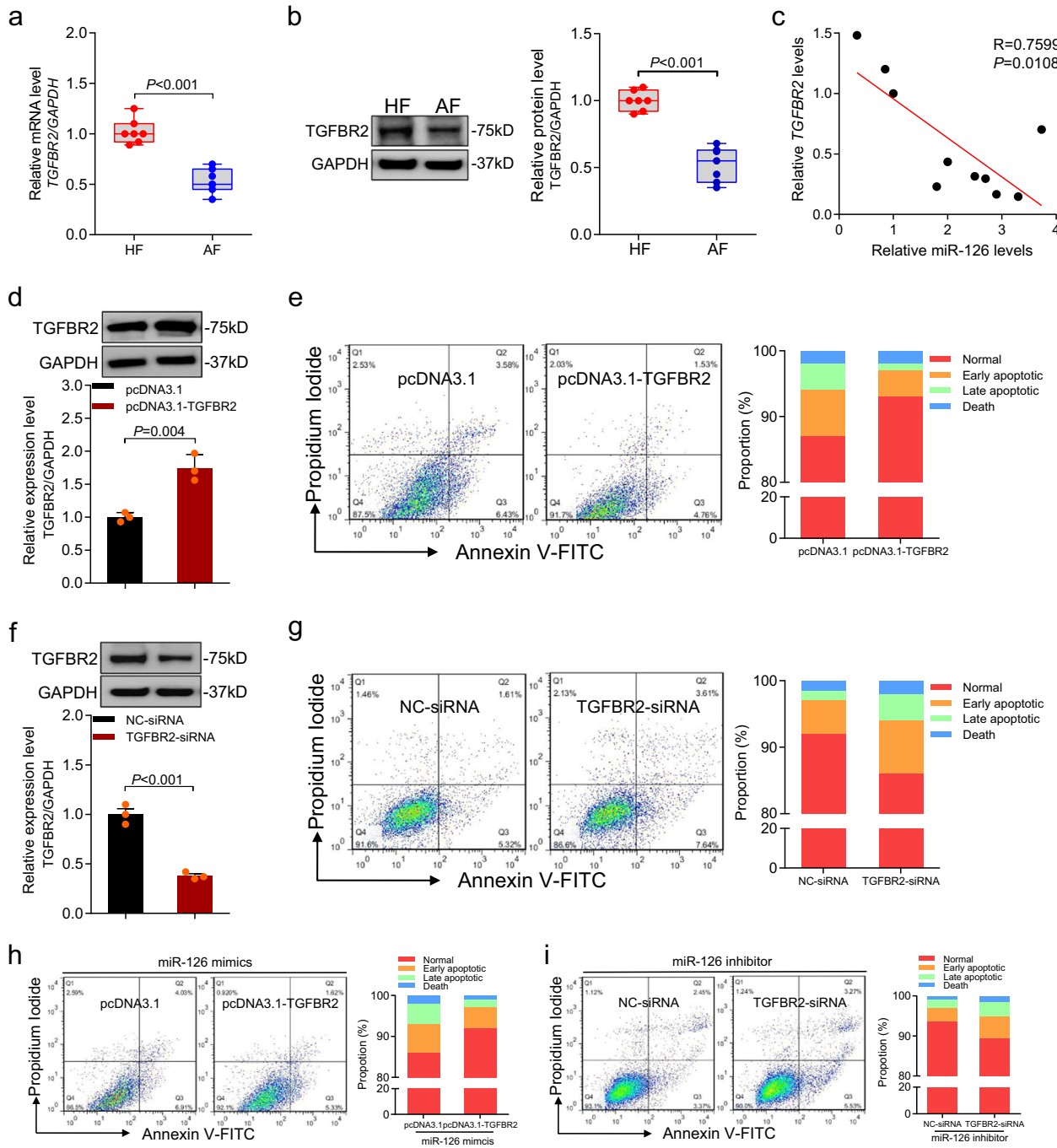

**Fig. 7 miR-126 enhances porcine granulosa cell apoptosis via targeting TGFBR2. a, b** qRT-PCR and western blotting showing the TGFBR2 mRNA (**a**) and protein (**b**) levels in granulosa cells from healthy follicles (HF) and atretic follicles (AF), n = 7 per group. **c** The relationship between miR-126 and TGFBR2 levels in the follicles of porcine ovary (n = 10). **d, e** Granulosa cells were transfected with pcDNA3.1-TGFBR2, the expression levels of TGFBR2 were detected by western blotting (**d**), and cell apoptosis was analyzed by FACS (**e**). **f, g** Granulosa cells were transfected with TGFBR2-siRNA, TGFBR2 protein levels were detected by western blotting (**f**), and the apoptosis rate was analyzed by FACS (**g**). **h, i** Granulosa cells were co-transfected with miR-126 mimics and pcDNA3.1-TGFBR2 (**h**) or miR-126 inhibitor and TGFBR2-siRNA (**i**), and cell apoptosis rate was determined by FACS. Data in **d, g** are shown as mean ± S.E.M. with three independent experiments. P values were calculated by a two-sided Student's t test.

decreased after treatment with *NORFA*-specific siRNA (Fig. 8i). In addition, we noticed that *NORFA* silencing suppressed the sensitivity of granulosa cells to TGF-β1, which could be restored by overexpression of *NORFA* (Supplementary Fig. 6a–c). Besides, we also observed that miR-126 decreased p-SMAD3 levels (Fig. 8j), whereas p-SMAD3 levels was upregulated after treatment with miR-126 inhibitor (Fig. 8k). Furthermore, miR-

126 suppressed *NORFA*-induced p-SMAD3 levels (Fig. 8l). On the other hand, miR-126 inhibitors had the ability to recover the levels of p-SMAD3, which was downregulated by *NORFA* silencing (Fig. 8m). However, pcDNA3.1-NORFA with the mutation type of miR-126-binding site (NORFA-mut^OE) had no effect on the expression levels of miR-126, TGFBR2, and p-SMAD3 in granulosa cells (Supplementary Fig. 7a–c). These

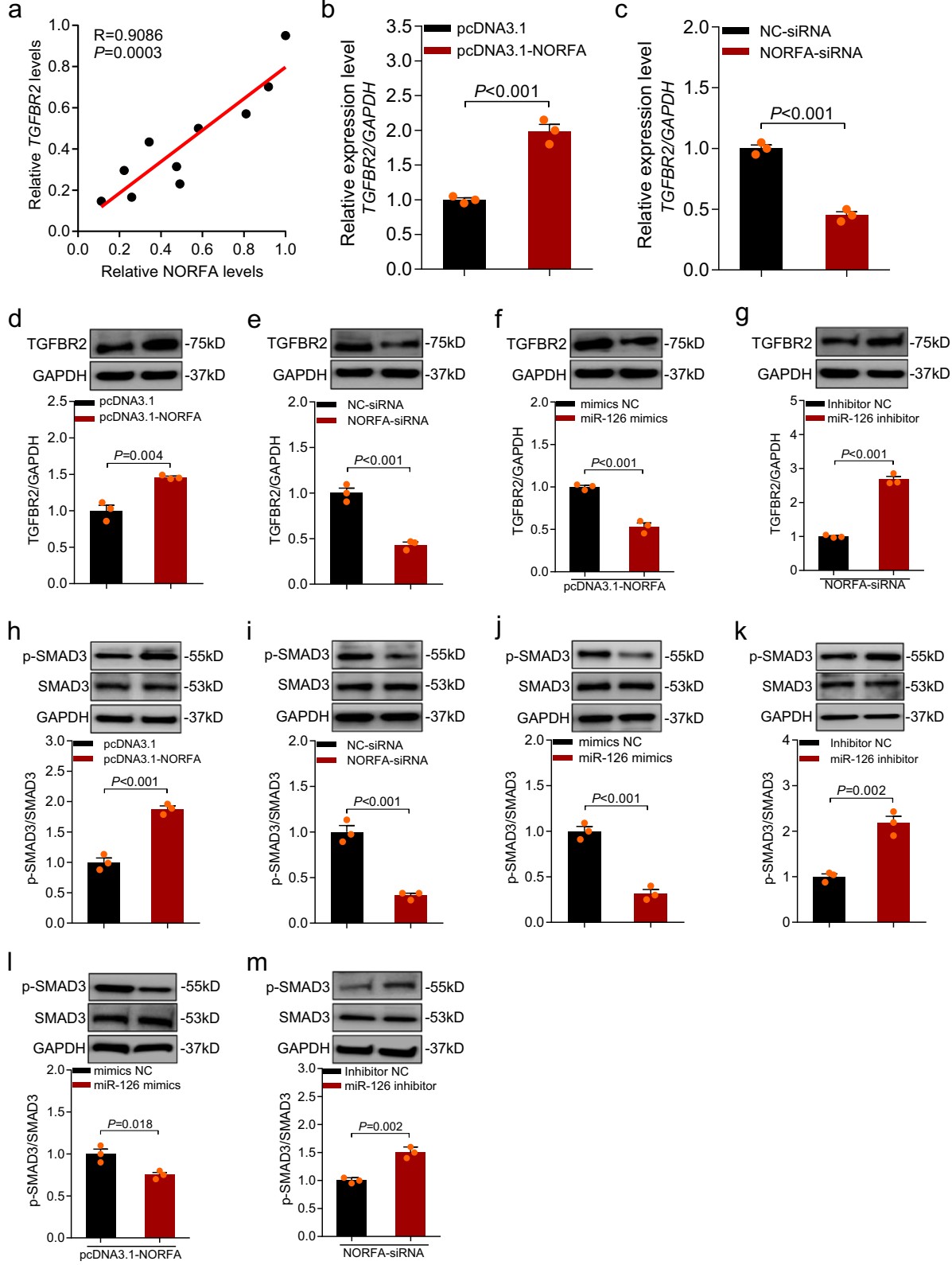

above-mentioned observations demonstrate that miR-126–TGFBR2 axis is involved in the regulation of *NORFA* to TGF-β signaling pathway in porcine granulosa cells.

**19-bp duplication in promoter influences *NORFA* expression.** To investigate the relationship between *NORFA* and pig

prolificacy, variants of *NORFA* gene in three pig breeds, Erhualian (known as one of the most prolific pig breeds in the world), Landrace, and Yorkshire (two famous western pig breeds), were screened and genotyped by individual sequencing. Five point mutations (g.-12C>T in promoter, g.249A>G, g.269A>G, g.290G>A, and g.380G>C in intron 1) and a 19-bp duplication (g.-1223-GCC AGG GCG GCA GTC CCG G-g.-1193) were

**Fig. 8 NORFA controls TGFBR2 and TGF-β signaling pathway through affecting miR-126 in porcine granulosa cells. a** Relationship between NORFA and TGFBR2 expression levels in the follicles of porcine ovary was detected (n = 10). **b–e** Granulosa cells were transfected with pcDNA3.1-*NORFA* or *NORFA*-siRNA, and TGFBR2 mRNA (**b**, **c**) and protein (**d**, **e**) levels were determined by qRT-PCR and western blotting, respectively. **f**, **g** miR-126 mediated NORFA regulation of TGFBR2 protein levels in granulosa cells. **h**, **i** The protein levels of p-SMAD3 and SMAD3 were measured after *NORFA* overexpression (**h**) or silencing (**i**). **j**, **k** The protein levels of p-SMAD3 and SMAD3 were measured after miR-126 overexpression (**j**) or silencing (**k**). **l**, **m** The protein levels of p-SMAD3 and SMAD3 in porcine granulosa cells were detected after treatment with miR-126 and NORFA. Data in **b–m** are shown as mean ± S.E.M. from three independent experiments. Relative protein levels were calculated and shown below are immunoblots. *P* values were calculated by a two-tailed Student's *t* test.

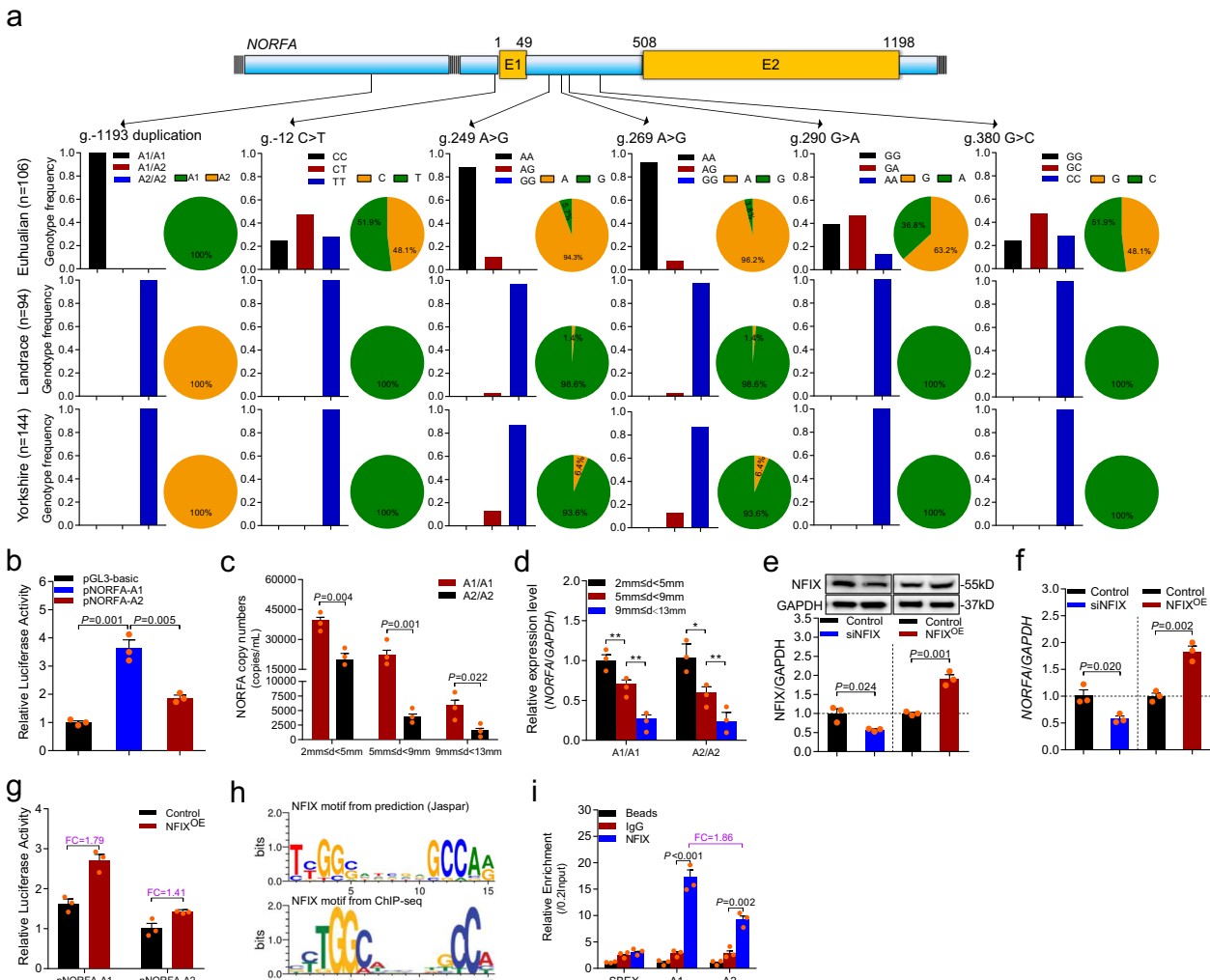

**Fig. 9 Breed-specific 19-bp duplication affects *NORFA* transcription by altering NFIX recruitment. a** Polymorphisms of the *NORFA* gene in Erhualian (n = 106), Landrace (n = 94), and Yorkshire (n = 144) sows. Schematic showing the *NORFA* gene and the location of 5 SNPs and a 19-bp duplication. Genotype and allele frequency were calculated and are represented with columns and pie diagrams, respectively. **b** The luciferase activities of *NORFA* promoter containing 19-bp duplication alleles were detected. **c**, **d** The copy number (**c**) and expression levels (**d**) of *NORFA* in granulosa cells (different sizes) from A1/A1 and A2/A2 genotype sows (n = 3) were detected by qRT-PCR. **e** NFIX protein levels in granulosa cells after overexpression or knockdown of NFIX were detected by western blotting. **f** The expression levels of *NORFA* in granulosa cells after NFIX was overexpressed or silenced were measured by qRT-PCR. **g** The effects of NFIX on the luciferase activities of *NORFA* promoter containing 19-bp duplication alleles. FC indicates fold change. **h** Analysis of NFIX-binding motif by Jaspar prediction tools (upper) and ChIP-seq data (lower). **i** ChIP-qPCR assay. The binding enrichment of NFIX on *NORFA* promoter containing 19-bp duplication alleles were detected by ChIP-qPCR assay. FC means fold change. Beads indicates no antibody, IgG was used as negative control, and the enrichment was normalized to 0.2 input. Data in **b–g** and **i** are represented as mean ± S.E.M. with three independent experiments. *P* values were calculated by a two-tailed Student's *t* test. *P < 0.05 and **P < 0.01.

identified within the *NORFA* gene (n = 344) (Fig. 9a; Supplementary Fig. 8). Notably, the 19-bp duplication located in the core promoter region of *NORFA* is a breed-specific variant, of which two copies (A1) only occurs in Erhualian pigs, while one copy (A2) exists only in Landrace and Yorkshire sows (Fig. 9a; Supplementary Fig. 9), suggesting that this variant may be involved in sow prolificacy. Luciferase assay showed that A1-type promoter has higher transcriptional activity than A2-type promoter (Fig. 9b). Similarly, the expression levels of *NORFA* in granulosa cells from A1/A1 ovarian follicles (2 ≤ d < 5 mm, 5 ≤ d < 9 mm, and 9 ≤ d < 13 mm) are higher than that in granulosa cells from A2/A2 sows, respectively (Fig. 9c, d). All these data

indicate that 19-bp duplication in *NORFA* promoter strongly influences *NORFA* level by regulating its transcription activity.

Prediction of binding sites for transcription factors revealed that five additional binding motifs for transcription factors including NFIX, E2F4, E2F6, GABPA, and PAX2 were detected in A1-type promoter region (Supplementary Fig. 10). With the highest prediction score, transcription factor NFIX was chosen for further investigation. We found that expression level of *NORFA* in granulosa cells was dramatically suppressed after knockdown of NFIX, whereas enhanced by ectopic expression of NFIX (Fig. 9e, f), revealing that NFIX is a transcription activator of *NORFA* in granulosa cells. Furthermore, we noticed that both A1 and A2 promoter activities were increased after NFIX overexpression, but the increased activity of A1 promoter induced by NFIX was higher than that of A2 promoter (Fig. 9g), indicating that 19-bp duplication influences NFIX regulation of *NORFA* transcription activity. In addition, chromatin Immunoprecipitation (ChIP)-qPCR assay confirmed that NFIX directly bind to NFIX-binding element (NBE) motifs in both A1- and A2-type promoters, but A1-type promoter could recruit more NFIX in comparison to A2-type promoter (Fig. 9h, i). Together, all data demonstrate that the 19-bp duplication influences *NORFA* transcription by altering the sensitivity of *NORFA* promoter to transcription factor NFIX.

### *NORFA* mediated NFIX inhibition of granulosa cell apoptosis.

We noticed that NFIX mRNA level in granulosa cells from healthy follicles (HFs) was higher than that from atretic follicles (AFs; Supplementary Fig. 11), indicating that NFIX is involved in porcine follicular atresia. Fluorescence-activated cell sorting (FACS) showed that the apoptosis rate of porcine granulosa cells was increased after NFIX silencing (Fig. 10a), whereas ectopic expression of NFIX suppressed cell apoptosis (Fig. 10b), suggesting that, like its target *NORFA*, NFIX is also an inhibitor of granulosa cell apoptosis and follicular atresia in porcine ovary.

We next investigated whether NFIX inhibits granulosa cell apoptosis by enhancing *NORFA*. As expected, NORFA overexpression could restore NFIX-siRNA-induced upregulation of cell apoptosis rate (Fig. 10a), while knockdown of *NORFA* could suppress NFIX-induced downregulation of cell apoptosis rate (Fig. 10b), demonstrating that *NORFA* mediated the inhibition of NFIX to porcine granulosa cell apoptosis. Besides, we also found that TGFBR2 protein level and p-SMAD3 activity were inhibited after knockdown of NFIX, which could be abolished by ectopic expression of *NORFA* (Fig. 10c). Opposite results were observed after transfected with pcDNA3.1-NFIX and *NORFA*-siRNA (Fig. 10d). All the data demonstrate that NORFA mediates the regulation of NFIX to porcine granulosa cell apoptosis and the activity of TGF-β signaling pathway.

### Discussion

Atresia and degeneration is the ultimate fate of most ovarian follicles (>99%) in mammalian ovary, only <1% of follicles undergo ovulation. Therefore, the study of the mechanism of follicular atresia is very important for decreasing the follicular atresia, increasing ovulation rate, and female fertility. It is now known that follicular atresia is triggered by granulosa cell apoptosis[32], which is thought to be involved in the death ligand receptor-induced and mitochondria-controlled apoptotic pathways. Previous studies have shown that follicular atresia and granulosa cell apoptosis were regulated by ncRNAs, especially by miRNAs such as miR-125b[33] and miR-1275[34]. However, lncRNA regulation of follicular atresia and granulosa cell apoptosis remains unclear. In this study, we proved that *NORFA*, a lincRNA, controls porcine granulosa cell apoptosis,

demonstrating that lincRNA is involved in follicular atresia of mammalian ovary. Although several lncRNAs were recently identified in mammalian ovary or ovarian cells including oocytes[35], somatic cells such as corpus luteum (CL) cells[36,37], and cumulus–oocyte complexes[38], knowledge of the role of lncRNAs in reproduction is still largely unknown. *Neat1* is one of the few lncRNA whose functions in mammalian ovary have been validated[27]. *Neat1* knockout (KO) mice are discovered to have prominent paraspeckle formation, dysfunction of the CL, low progesterone levels in serum, ovarian defects, and decreased fertility, suggesting that *Neat1* is required for CL formation and the establishment of pregnancy[27]. Notably, *NORFA* is a pig-specific lincRNA and may be the breakthrough for us to reveal the mechanism that lincRNA regulates female reproduction in pigs. Further studies are needed to investigate the role of *NORFA* in regulating other functions of granulosa cells (e.g., steroidogenesis), follicular atresia in vivo, and reproductive performance such as litter size in pigs.

Our mechanism studies underlying *NORFA* regulation of porcine granulosa cell apoptosis and follicular atresia have revealed that *NORFA* acts as a ceRNA of nearby miR-126 gene, which was upregulated during ovarian follicular atresia. The ceRNA hypothesis posits that specific RNAs, such as mRNAs, pseudogene transcripts, circular RNAs, and lncRNAs, can impair miRNA activity through acting as molecular sponges for endogenous miRNAs, thereby enhancing miRNA target expression[39,40]. Acting as a ceRNA is the main function mode for lncRNAs containing the same MREs with targets[41,42]. For example, *TGFB2-OT1* sponges miR-3960, miR-4488, and miR-4459 to control autophagy and inflammation of vascular endothelial cells[43]. In this study, miR-126 is an intragenic miRNA transcribed from the intron of *EGFL7*[44], which is a nearby coding gene of *NORFA*. Not unexpected, we found that miR-126 and *EGFL7* were oppositely regulated by *NORFA*. However, some issues such as how *NORFA* regulates the expression of nearby coding genes and the molecular mechanism of how *NORFA* controls granulosa cell apoptosis and follicular atresia through these nearby coding genes are very interesting questions for future investigation.

As an intronic miRNA, miR-126 has been shown to regulate multiple important cell biological processes, such as physiological processes including cell proliferation, apoptosis, differentiation, adhesion, and intercellular interaction[45,46]; and pathological processes including cell cycle, differentiation, and self-renewal of human acute myeloid leukemia stem cells[47]; and cell proliferation, survival, apoptosis, migration, invasion, motility, and metastasis of cancer cells[44,46,48]. In the present study, we have shown that miR-126 promotes porcine granulosa cell apoptosis and follicular atresia, demonstrating that miR-126 is required for the functions of granulosa cells in female mammals. In normal cells, KO mouse model showed that plasmacytoid dendritic cell apoptosis rate is increased in Mir126$^{-/-}$ mice, and miR-126 regulates the expression of innate response genes (e.g., *Tlr7*, *Tlr9*, *Nfkb1*, and *Vegfr2*) and directly targets the mammalian target of rapamycin pathway[49]. In pathological cells, miR-126 widely regulated cell apoptosis of various cancer cells, such as mesothelioma cells[50] and cervical cancer cells[51].

Increasing studies have demonstrated that miRNAs directly interact with TGF-β signaling pathway to control the state and function of granulosa cells. Meanwhile, few reports demonstrate that lncRNAs also directly interact with TGF-β signaling pathway, but hundreds of TGF-β-induced lncRNAs have been identified, mainly in cancer cells[52]. Here we showed that *NORFA* governs TGF-β signaling pathway-mediated porcine granulosa cell apoptosis and follicular atresia by interacting with miR-126. TGF-β-activated lnc-*ATB* is involved in TGF-β signaling

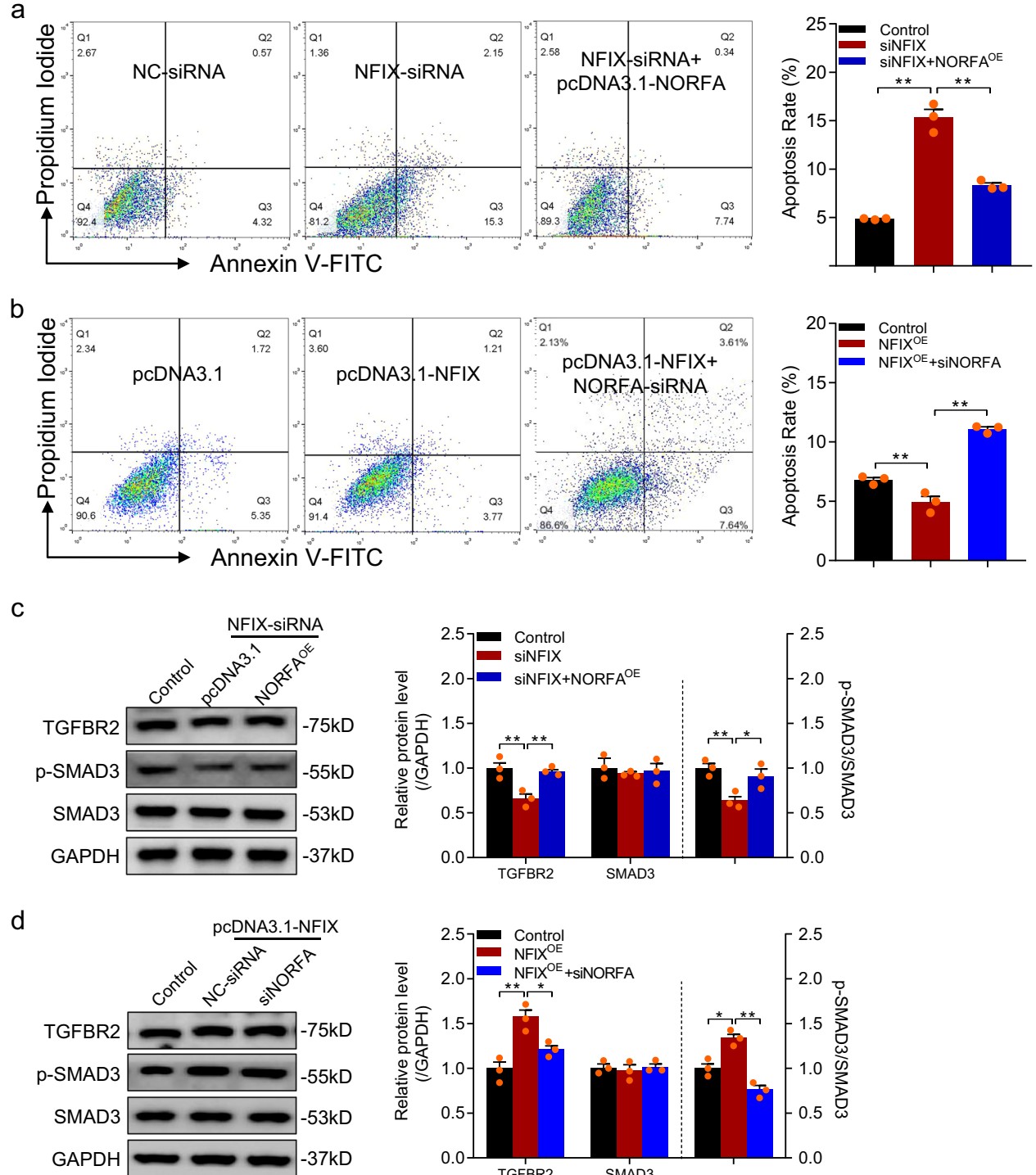

**Fig. 10 NFIX/NORFA axis regulates porcine granulosa cell apoptosis by activating TGF-β signaling pathway. a** The apoptosis rate of porcine granulosa cells co-transfected with NFIX-siRNA and pcDNA3.1-*NORFA* were determined by FACS. **b** The apoptosis rate of porcine granulosa cells co-transfected with pcDNA3.1-NFIX and *NORFA*-siRNA were detected by FACS. **c** The protein levels of TGFBR2, p-SMAD3, and SMAD3 in granulosa cells after treatment with NFIX-siRNA and pcDNA3.1-*NORFA* were measured by western blotting. **d** The protein levels of TGFBR2, p-SMAD3, and SMAD3 in granulosa cells after treatment with pcDNA3.1-NFIX and *NORFA*-siRNA were analyzed by western blotting. Data are shown as mean ± S.E.M. with three independent experiments. *P* values were calculated by a two-tailed Student's *t* test. *$P < 0.05$ and **$P < 0.01$.

pathway, which interacts with miRNAs to govern TGF-β signaling pathway in various cells, including enhancement of TGFBR2 and SMAD2 by inhibiting the endogenous miR-425-5p in hepatic stellate cells[53] and directly binding to miR-141-3p in gastric cancer cells[54]. In addition, TGF-β recruits the Smad proteins and the CCCTC-binding factor complexes on the *H19*

imprinting control region chromatin, and the complexes could function as insulator in *cis* and regulator of transcription and replication in *trans*[55].

In recent years, lots of important trait-associated variants within lncRNAs have been identified using high-throughput methods such as genome-wide association studies (GWAS) and whole-genome

sequencing[56–58]. Most of these variants are disease-associated variants in human and model animals such as mouse[59,60]. In human fetal tissues, tissue m6A regions have been shown to be enriched for GWAS variants; lipid trait- and metabolic trait-associated variants were most strongly enriched in liver m6A regions: the variant rs7305618 that associated with type 2 diabetes in lncRNA *HNF1A-AS1* is one of them[58]. Several variants have been identified as biomarkers for cancer and other diseases, such as variant rs11672691 in lncRNA *PCAT19* for prostate cancer[61] and variant rs1015164 in lncRNA *CCR5AS* for human immunodeficiency virus disease[60]. Here we identified, for the first time to our knowledge, a sow prolificacy-associated variant, 19-bp duplication in the promoter region of lincRNA *NORFA*. In domestic animals, only a few lncRNA variants associated with important economic traits were found, including variants rs325797437, rs344501106, rs81286029, and rs318656749 in lncRNA *MEG3* associated with porcine meat production traits[62], a variant g.1263T>A in lncRNA *ADNCR* associated with bovine growth traits[63], and a variant in lincRNA *GDNF-AS* associated with canine insensitivity to pain[64]. Overall, based on the distribution of the 19-bp duplication in pig breeds with different fertility and the expression levels of *NORFA* in granulosa cells from individuals with different genotypes, we identified for the first time that a candidate lincRNA for sow prolificacy, providing insights into the molecular mechanism of the high prolificacy of Erhualian pigs.

Although large numbers of important trait-associated variants have been identified in mammals, mainly humans, the functional explanations and mechanism underlying regulation of gene expression for the majority of variants remain unknown[58,60]. In this study, we showed that 19-bp duplication in *NORFA* promoter creates a binding site for transcription factor NFIX, which strongly enhances transcription activity of *NORFA* in granulosa cells and transcription level in follicles through recruiting more transcription factor NFIX to the promoter region of *NORFA*. Furthermore, NFIX is involved in porcine follicular atresia and granulosa cell apoptosis via NORFA and NORFA-mediated TGF-β signaling pathway. NFIX belongs to nuclear factor one (NFI) family of transcription factors that consists of four members (the other three are NFIA, NFIB, and NFIC)[65]. In mammals, NFIX is widely expressed in various tissues and organs, which plays a critical role in development and disease[66,67]. As a transcription factor, NFIX has a bi-phase function in regulating target expression by directly binding a consensus sequence TTGGCN$_5$GCCAA or 1/2 sites (i.e., TTGGC or GCCAA) of target promoter and then controlling the target-mediated cell functions[65,67]. Interestingly, in various cell functions, NFIX has been shown to be crosstalk with TGF-β signaling pathway. On one hand, TGF-β1 controls NFIX expression in myogenic differentiation[68] or in cellular senescence[69]. On the other hand, NFIX can form a nuclear complex of NFIX1/Smad4 to contribute to contact inhibition[70] and oxidative stress in cellular senescence[69]. These observations not only uncover the function of 19-bp duplication and its mechanism underlying regulation of *NORFA* transcription but also define the function of NFIX in the ovary and establish a link between NFIX and the TGF signaling pathway in granulosa cells.

In conclusion, we identified and characterized a pig-specific lincRNA, *NORFA*, which is highly expressed in ovary and involved in ovarian follicular atresia. Our findings represent a clear mechanism that *NORFA* controls ovarian follicular atresia: *NORFA* sponges endogenous miR-126 in porcine granulosa cells and then prevents miR-126 binding to the target *TGFBR2* 3′-UTR, which further releases TGFBR2 and activates TGF-β signaling pathway, thereby inhibiting cell apoptosis and follicular atresia. Importantly, we also identify a sow prolificacy-associated variant, 19-bp duplication in the promoter region of lincRNA *NORFA*. This variant enhances *NORFA* transcription through recruiting more transcription factor NFIX to the promoter region of *NORFA*, which is involved in follicular atresia and granulosa cell apoptosis. Overall, our findings provide insights into the molecular mechanism of follicular atresia and the high prolificacy of Erhualian pigs and a target for improving the reproductive performance of other pig breeds (e.g., Landrace and Yorkshire) by using marker-assisted selection and gene editing technology.

## Methods

**Compliance with ethical regulations**. All animal experiments were approved by Animal Ethics Committee at Nanjing Agricultural University, China (SYXK (Su) 2015-0656) and performed in accordance with the Regulations for the Administration of Affairs Concerning Experimental Animals (No. 2 of the State Science and Technology Commission, 11/14/1988).

**Cell culture and treatment**. Porcine granulosa cells from healthy ovarian follicles (3–5 mm diameter) were collected and cultured as previously described[24]. HEK293T cells were maintained in Dulbecco's modified Eagle's medium (Gibco) supplemented with 10% fetal bovine serum (FBS, Invitrogen) at 37 °C with a humidified air atmosphere containing 5% $CO_2$. For transfection, granulosa cells and HEK293T cells were seeded into 6- or 12-well plates and Lipofectamine™ 3000 Transfection Reagent (#L300015, Life Technologies, CA92008, USA) was used for transfection. For TGF-β treatment, cells were treated with TGF-β1 (10 ng/mL, R&D System) followed by maintaining in medium without FBS for 12 h. The oligonucleotides used were synthesized by GenePharma (Shanghai, China) and listed in Supplementary Table 1.

**Follicular classification**. Follicles were isolated from ovaries and classified as HFs and AFs according to the morphological identification, granulosa cell density, and progesterone/estradiol (P4/E2) ratio. Briefly, ovarian follicles with a red color having small blood vessels, low antral granulosa cell density (≤2500/μL), and a low P4/E2 ratio (≤2.0) were categorized as HFs. While, the transparent colorless follicles with high antral granulosa cell density (>2500/μL) and a high P4/E2 ratio (>2.0) were categorized as AFs. Only follicles with morphology in accordance with P4/E2 values were selected for further investigation. For detection of the copy numbers of *NORFA* in granulosa cells, ovarian follicles of different size ($2 ≤ d < 5$ mm, $5 ≤ d < 9$ mm and $9 ≤ d < 13$ mm) were isolated.

**RNA isolation and quantitative real-time PCR**. Total RNA was isolated from granulosa cells using the High Purity Total RNA Extraction Kit (#RP5511, Bioteke, China) and reverse transcribed into cDNA by PrimeScript™ RT Master Mix (#RRO36A, TaKaRa). qRT-PCR was performed with AceQ® qPCR SYBR® Green Master Mix (#Q111-02, Vazyme Biotech co., ltd.), and the relative expression levels were calculated with $2^{-ΔΔCt}$ method. *GAPDH* was used as endogenous control for *NORFA* and coding genes. To measure the miR-126 expression level, miR-126 stem-loop primer was designed to synthesize cDNA and the specific primers of miR-126 were designed for qRT-PCR. *U6* small nuclear RNA acted as an internal control. The copy numbers of *NORFA* in granulosa cells from Erhualian purebred sows (A1/A1) and Yorksire cross-bred sows (A2/A2) were measured by copy number reference assay with qPCR. The standard curve was produced by using a 10-fold serial dilution of purified *NORFA* standards. Besides, reaction efficiencies were measured through establishing single and multiplex standard curves. Primers used here are listed in Supplementary Table 2.

**Bioinformatics analysis**. The genomic location, conservation, and GC percent of *NORFA* were obtained from ensembl database (http://www.ensembl.org/index.html). The coding potential of NORFA was analyzed by PhyloCSF and CPAT (http://lilab.research.bcm.edu/cpat/). The MREs of ssc-miR-126 in NORFA were predicted by starbase v2.0 (http://starbase.sysu.edu.cn/mirLncRNA) and miRcode (http://www.mircode.org/). The mature sequences of miR-126 from different species were obtained from miRBase (http://www.mirbase.org), and their genomic locations were analyzed by NCBI database (https://www.ncbi.nlm.nih.gov/). The candidate miR-126 target genes were analyzed using four different algorithms, microRNA.org (http://www.microrna.org/), TargetScan (http://www.targetscan.org/), PITA (http://genie.weizmann.ac.il/pubs/mir07), and TarBase (http://diana.cslab.ece.ntua.gr/tarbase/). Venn diagram indicating the intersected genes was generated using a Draw Venn Diagram online tool (http://bioinformatics.psb.ugent.be/webtools/Venn/). DAVID (Database for Annotation, Visualization and Integrated Discovery, version 6.7, https://david.ncifcrf.gov/) was used for KEGG pathways analysis of miR-126-target genes. A significance level of $P < 0.05$ and an enrichment score >2 were set as the thresholds. RNAhybrid was used to measure minimum free energy (http://bibiserv.techfak.uni-bielefeld.de/rna-hybrid/). The binding motifs of different transcription factors within *NORFA* promoter were predicted by JASPAR (http://jaspar.genereg.net/). Heatmap was generated by using R package (heatmap).

**Rapid amplification of cDNA ends**. The 5′ and 3′ ends of NORFA were identified by using the SMARTer® RACE 5′/3′ kits (#634858, Clontech Laboratories, CA94043, USA). Briefly, 500 ng total RNA from porcine granulosa cells was used as template to synthesize 5′- and 3′-RACE-Ready first-strand cDNA. Specific primers were designed for RACE following the instructions. The 5′-RACE reversed primer was: 5′-GAT TAC GCC AAG CTT GTC GTC CTG CGA CGT CTC CGC AG-3′. The 3′-RACE forward primer was: 5′-GAT TAC GCC AAG CTT TGG TCT CCC AAA GGC CTG-3′. Amplification products were identified by electrophoresing on 2% agarose gel and cloned into T3 vectors for sequencing to get the full-length sequence of NORFA. The full length of porcine NORFA has been submitted to the NCBI Genbank database (#MK879596.1).

**Histology analysis**. The porcine ovaries were fixed in 10% formalin in phosphate-buffered saline (PBS). The fixed specimens were dehydrated through an ethanol series (70%, 80%, 90%, and 100%), cleared in xylene, and embedded in paraffin according to standard techniques. Ovarian slices from the embedded specimens were cut into 4-μm-thick sections, stained with hematoxylin and eosin, and analyzed by microscopic examination.

**RNA fluorescence in situ hybridization**. RNA FISH for NORFA was performed by Goodbio technology (Wuhan, China). Briefly, porcine ovaries were collected and immediately frozen in liquid nitrogen. The sliced samples were postfixed in 4% paraformaldehyde, acetylated in 1% triethanolamine and 0.25% acetic anhydride, prehybridized, and hybridized at 65 °C using the following anti-sense probes: 5′-TCG TCC TGC GAC GTC TCC GCA GGG TTC CTG CTG CCG AGG GGA GCT TCA CCT GCA GCA CTG CAC CCC TCT TTC TTC AGG GTA AAA TGA GGA AGT CGC CTT CTG CGG TGA-3′. Nucleus was stained with 4,6-diamidino-2-phenylindole dye (blue). To detect the location of NORFA in porcine granulosa cells, cells were rinsed briefly in cold PBS and fixed with 4% formaldehyde for 20 min at room temperature. Then cells were permeabilized in PBS (0.5% Triton X-100) on ice for 5 min, and hybridization was performed using the probes above. Images were obtained from fluorescence microscopy using a Nikon Eclipse 80i microscope equipped with a Nikon DS-2 digital camera.

**Subcellular localization**. Porcine granulosa cells were collected for nuclear and cytoplasmic extraction using the method described by Wei et al.[71] with slight modifications. Briefly, granulosa cells were lysed in cold lysis buffer (10 mM Tris, 0.1 mM EDTA, 1 mM NP-40, and 1 mM RNase inhibitor) for 5 min on ice and then centrifuged at $10,000 \times g$ for 3 min at 4 °C. The supernatant cytoplasmic extract was immediately frozen (−80 °C) for subsequent analysis. The nuclear pellet was resuspended on ice with DEPC water with 1 mM RNase Inhibitor for 3 min and centrifuged at $10,000 \times g$ for 3 min. The supernatant nuclear extract was removed and the remainder was frozen (−80 °C) for subsequent analysis.

**Western blotting**. Western blotting assays were performed as described previously[31]. The primary antibodies used here are anti-TGFBR2 (Sangon Biotech, #D155818, 1:1000), anti-SMAD3 (Sangon Biotech, #D155234, 1:1000), anti-p-SMAD3 (Sangon Biotech, #D155153, 1:1000), anti-NFIX (Affinity, #DF3250, 1:1000), and anti-GAPDH (ORIGENE, #TA802519, 1:3000). GAPDH was measured as an internal control.

**Apoptosis analysis**. Apoptosis Detection Kit (#A211-01, Vazyme biotech co., ltd, China) was used to measure cell apoptosis rate. Briefly, 20,000 cells were stained with Annexin V-fluorescein isothiocyanate/propidium iodide and then sorted by FACS using a cell counting machine (Becton Dickinson). The apoptosis rate was analyzed using the Flowjo software (TreeStar) and calculated using the following equation: (number of cells in the right upper quadrant + number of cells in the right lower quadrant)/(total number of cells)[24].

**Plasmids**. To identify the MRE of miR-126 in NORFA or *TGFBR2* 3′-UTR, porcine NORFA or *TGFBR2* 3′-UTR fragments containing potential miR-126 MRE motif were isolated and inserted into pmirGLO Dual-luciferase miRNA Target Expression Vector between NheI and XhoI enzyme sites (#E1330, Promega). For NORFA overexpression vector construction, full-length sequence of porcine NORFA was synthesized and cloned into pcDNA3.1 expression vector (#V790-20, Invitrogen) between NheI and XhoI enzyme sites. To detect the effects of NFIX on the activity of NORFA promoter, porcine NORFA promoters with −1193 ins allele and del allele (which contains 2 and 1 NFIX-binding motif, respectively) were cloned into pGL3-Basic luciferase reporter vector (#E1751, Promega) between KpnI and XhoI enzyme sites. To generate the NFIX expression vector, full-length coding sequence of porcine NFIX was amplified by using total RNA extracted from porcine granulosa cells and subcloned into pcDNA3.1 expression vector within KpnI and XhoI enzyme sites. For RNA pull-down assays, pSPT19-NORFA and pSPT19-NORFA-mut (miR-126) were designed and synthesized by GenScript Inc. Mutant plasmids were generated by using a TaKaRa MutanBEST Kit (#R401, TaKaRa) according to the manufacturer's protocol. All of the recombinant plasmids were

confirmed by Sanger sequencing. The primers used here are shown in Supplementary Table 3.

**Dual luciferase assay**. After transfection for 24 h, cells were collected and luciferase activities of each sample were detected by using Dual-Luciferase Reporter Assay System (#E1910, Promega) according to the manufacturer's protocol. GLOMAX detection system (Promega) was conducted to measure the luciferase activity in cell lysates. Relative luciferase activity of each sample was calculated as the ratio of *Firefly/Renilla*.

**RNA pull-down assay**. NORFA and NORFA-mut (miR-126) were transcribed from vector pSPT19-NORFA and pSPT19-NORFA-mut (miR-126) in vitro, respectively. After biotinylated modification with the Biotin RNA Labeling Mix (#No.11685597910, Roche) and T7 RNA polymerase (#EP0111, ThermoFisher Scientific), two single-stranded RNA transcripts were purified with an RNeasy Mini Kit (#74104, Qiagen, Valencia, USA). The purified biotinylated transcripts (5 μg) were incubated with 20 μg porcine granulosa cells total RNA for 4 h at room temperature. Streptavidin magnetic beads (#LSKMAGT02, Merck Millipore) were used to isolate the biotin-NORFA/RNA complex according to the manufacturer's protocol. After isolation, the levels of target miRNAs present in the pull-down material were detected by qRT-PCR analysis.

**Chromatin immunoprecipitation**. ChIP assays were performed as previously described[31]. Briefly, Rabbit anti-NFIX (Affinity, #DF3250) antibody was used to pull down NFIX-DNA complex and a nonspecific antibody against IgG (Santa Cruz, #sc-2358) was used as an internal control, and the chromatin before immunoprecipitation was used as an input control. The specific PCR primers used to detect NFIX-binding motifs within NORFA promoter are listed in Supplementary Table 4. Each specific antibody ChIP sample was normalized to relative IgG antibody ChIP signals obtained from the same sample. ChIP-qPCR signals were calculated as fold enrichment of input control signals with technical triplicates.

**Animal selection, sample collection, and DNA extraction**. A total of 344 sows were randomly selected for NORFA single-nucleotide polymorphism (SNP) screening (106 Erhualian sows from Jiaoxi Erhualian Breeding Farm, Jiangsu Province, China; 144 Yorkshire sows from Yongkang Farming Technology Co., Ltd., Jiangsu Province, China; 94 Landrace sows from Huaiyin Breeding Farm, Jiangsu Province, China). Ear tissue sample (~2 g per sow) from each sow was collected in a cryotube and kept in 75% ethyl alcohol (v/v). All efforts were made to minimize animal suffering during ear tissue collection. The genomic DNA from each ear sample was extracted according to the standard phenol–chloroform method and the purified DNA was stored at −20 °C for further experiments.

**Polymorphism analysis**. Six pairs of primers were designed to analyze the polymorphisms of NORFA according to the porcine NORFA sequence. With the genomic DNA mentioned above as substrates, PCR was performed by using 2× Vazyme LAmp Master Mix (#P312-01, Vazyme biotech co., ltd, China), and the products were confirmed by Sanger sequencing. Subsequently, the variants were analyzed by using a DNA sequencing software Chromas. Linkage disequilibrium and haplotypes were calculated and identified by using SHEsis (http://analysis.bio-x.cn/myAnalysis.php) and Haploview 4.2 software with the SNP information of porcine NORFA gene. The primers used here are listed in Supplementary Table 5.

**Statistics and reproducibility**. All experiments were repeated at least three times and the data are presented as means ± S.E.M. Statistical analyses were performed using IBM SPSS Statistics v20.0 (SPSS Inc.). An unpaired two-tailed Student's $t$ test was used to evaluate the significance. Significance between conditions are denoted as $*P < 0.05$ and $**P < 0.01$. All the figures were built by the GraphPad Prism v7.0 software.

**Reporting summary**. Further information on research design is available in the Nature Research Reporting Summary linked to this article.

## Data availability

The majority of the data in this study are included in this published article. The DNA sequence of NORFA is available in the NCBI Genebank database (#MK879596.1). The source data underlying the graphs and charts presented in the main figures are shown as Supplementary Data 1. The original images of gels and blots are shown in Supplementary Fig. 12. The newly constructed plasmids have been deposited into Addgene (deposit number: 77871). Other information supporting the findings of this study is available from the corresponding author on reasonable request.

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

## Acknowledgements
We are grateful to Zaohang Jiang for his excellent technical assistance and advice. This research was supported by the National Natural Science Foundation of China (31630072, 31902130), China Postdoctoral Science Foundation (2018M632321), and the Natural Science Foundation of Jiangsu Province (BK20180524).

## Author contributions
X.D. and Qifa Li conceived and designed the project. X.D. performed the experiments, L. L. and Qiqi Li assisted with experiments. X.D., L.Z., and Z.P. analyzed the data. X.D. wrote the manuscript with input from all other authors. All authors critically reviewed and approved the final manuscript.

## Competing interests
The authors declare no competing interests.
