## [Peer Review File · Communications Biology]

Reviewers' comments:

Reviewer #1 (Remarks to the Author):

Du X et al. identified novel lincRNA, NORFA from SMAD4-silenced porcine GCs. They found that NORFA has anti-apoptotic activity in GC by functioning as a ceRNA of miR-126, which has pro-apoptotic activity in GC. They also found that the pro-apoptotic activity of miR-126 is via the direct targeting of TGFβ2. They confirm that anti-apoptotic activity of NORFA is through the inhibition of miR-126 to target TGFβ2. In addition, they found that NORFA promoter has 19-bp duplication that is targeted by transcriptional activator, NF1X, which is highly expressed in healthy follicles. Based on these results, they concluded that NORFA functions as an inhibitor of granulosa cell apoptosis. Overall, the data is interesting and well designed to draw conclusion. However, I have 3-concerns that (1) some of subtitle is not matching with the data (2) The manuscript is too lengthy. I think that Figure 1-8 and 9-10 is different story. Is it necessary to combine these data all together in the manuscript? (3) Again, Discussion is too too lengthy. Discussion should be the discussion about the data in the manuscript.

Detailed comments are as follow,

- (1) line 87 and line 100: The subtitle and conclusion of this part needs to be edited. The data of figure 2 support that NORFA is essential for GC apoptosis but low expression of NORFA in follicular atresia (Fig. 1g, h) does not mean it is essential in follicular atresia.
- (2) line 96: What does it mean "NORFA is an anti-apoptotic epigenetic mediator"?
- (3) line 110: Please insert reference for "miR-126, an intronic miRNA transcript from EGFL7"
- (4) line 137: The subtitle "in vivo" is not appropriate.
- (5) Line 159 and line 176: The subtitle and conclusion of figure-5 is confusing. miR-126 induces GC apoptosis and NORFA inhibits pro-apoptotic activity of miR-126, right?
- (6) Line 199: Figure 6g and 6i, Western band seems to be saturated. Can we expect more clear western result by reducing the amount of total protein in the western blot analysis?
- (7) Line 229 and Line 253: The subtitle and conclusion of figure-8 is confusing. miR-126 inhibits the NORFA activity to enhance TGF-β signaling, right?
- (8) Fig 8-H, I, K: The difference of band intensity between control and test is marginal. Can we expect more clear western result by reducing the amount of total protein in the western blot analysis?
- (9) Figure-9 and -10: I think that the conclusion of the manuscript is solid with figure 1-8.
- (10) Discussion is too lengthy.

Reviewer #2 (Remarks to the Author):

Comment and suggestions for authors:

The manuscript "NORFA, a novel candidate lincRNA for sow fertility, inhibits granulosa cell apoptosis" describes the role of a lincRNA, NORFA in porcine granulosa cell (GC) apoptosis, follicular atresia, and sow fertility. The study sort to examine if NORFA was involved in GC apoptosis and follicular atresia and to determine the actual role it plays and its mechanism of action. The study results and discussion propose that NORFA sponges endogenous miR-126 in porcine GCs and prevent its binding to the 3'UTR of TGFβ2, releasing TGFβ2 to inhibit GC apoptosis and follicular atresia. The study also identified a 19-bp duplication in the promoter region of NORFA which is a sow prolificacy-associated variant that recruits the transcription factor NF1X to enhance NORFA transcription and regulation of GC apoptosis and follicular atresia.

The study was thoroughly conducted with adequate samples, replicates, controls. The aim is clear and technically sound methodology was used to arrive at the conclusion. Sufficient data have been provided to support the claims of the study and the data is made available. The discussion is elaborate

and the conclusion is drawn from the purpose of the study and the results obtained from the experiments. Appropriate references have been cited when necessary in most cases for a comprehensive understanding of the study.

The manuscript is written in standard English.

However, the following revisions need to be made.

Specific comments

Revise the following statements at abstract to make them meaningful and clear;

- The first sentence in the abstract (line 8) does not express any coherent idea. It seems to suggest that "lincRNAs have been implicated in healthy and disease conditions". Restructure the statement to reflect so, if that is what the sentence seeks to suggest.
- Line 15-16 (furthermore, the correlations among NORFA, miR-126 and TGFBR2 levels were validated in follicles. SUGGESTION: The correlation between NORFA, miR-126, and TGFBR2 levels in follicles was further validated)

The introduction or background of the study has no heading/subheading (line 22).

Cite reference for the ideas expressed in the statement at lines 35 to 37 of introduction.

At results section, line 108-109, the statement is ambiguous, restructure it to indicate the expression level of the four genes were increased.

The meaning of the statement at discussion (line 349-350) is not clear. Revise it to capture what you want to you want to express.

Minor suggestions

Below are some recommended suggestions for some words or statements in the manuscript:

Replace "we report a novel lincRNA" (line10) with (we report that a novel lincRNA),". Replace "Prevented" with "preventing" (line 13). "Are identified" with "have been identified" (line29-30). Put "are" between miRNAs and mainly (line 35). Replace "which" with "with" and "show" with "showing" (line 38). "To" with "and" (line 49). "It is little known about" should be replaced with "little is known of" (line 51).

At results;

Change "with highly expressed" to "which was highly expressed" (line 64). Replace "is" with "to be" and "to locate" with "to be located" (line 67). Put "the" before "genome" (line 70). Replace "the" with "a" (line 71)

Put "more" before "especially" (line 75). Put a comma after "that" (line 90). Put "our" after "all" (line 100). "comprising" is much more appropriate than "including" (line 107). Put a semicolon after "including" (line 108). Change "were decreased" to "showed decreased expression" (line 109-110). Put "the" after "and" (line 113).

Put a full stop after "GCs" and begin the next sentence with a capital (line115). Put "of the" before NORFA (line 121). Change "physically" to "physical" (line 127). Replace "not" with "no" (line131).

Rephrase this statement at lines 137-138; "due to the lack of the characterization of the gene encoding miR-126 in pig". SUGGESTION: change "the lack of" to "the unavailability of data or information on". Change "highly" to "high" (line 140). Put "is" before "consistent" (line 151).

Change "we next to analyse" to "next we analyzed" (line 159-160). Replace "positive" with "positively" (line 232). Do same on line 233. Change "detected" to "examined the" (line 242). Replace "which" with "was" (line252). Change "breeds" to "breed" (line 262). Put "of" in front of "which" (line 268). Change "only exists" to "exists only" (line 268). Change "investigate" to "investigation" (line 281).

At discussion;

Put "of" in front of "follicles" (line 318). Put "the" in front of "mechanism" (line 319). Replace "reproductive" with "reproduction" (line 333). Change "that the" to "whose" (line 334), "opposite" to "oppositely" (line 356). Put a semicolon after "including" (lines 362 and 364). Change "we showed" to "we have shown" (line 367). Replace "was" with "has been" (line 384). Change "have" to "have been" (line 402). Change "a biomarker" to "biomarkers", and "disease" to "diseases" (line403). Change

"identify" to "identified" (line 415). Change "functions" to "function" (line 432), and "direct" to "directly" (line 433). Remove "the" (line 436). Put "is" after "which" (line 452).

At method;

Remove "through" (line 510). Change "end" to "ends" (line 524). Put "was" before the semicolons on lines 529 and 530. Replace "as" with "by" (line 557). Change "presenting" to "present" (line 615). Replace "served" with "use" (line 620). Put "tissue" in front of "sample" (line 632).

Supplementary data

At supplementary figure 6, "schematic" should be "schematic diagram".

BEST WISHES.

Reviewer #3 (Remarks to the Author):

In this manuscript by Du et al., authors identified that pig-specific lncRNA NORFA can regulate granulosa cell apoptosis by acting as a sponge for miR-126, and further demonstrated that NORFA/miR-126 axis plays an important role in regulating GCs apoptosis through targeting TGFBR2. In the end, authors identified a pig-specific 19-bp duplication in NORFA promoter, which could regulate NORFA transcription by altering the recruitment of NFIX to the promoter of NORFA.

Overall, this study is very interesting and authors provided comprehensive experiments. However, the results were not solid enough to support the conclusions. Below are my major comments.

1, Authors stated this lncRNA is pig-specific, but did not provide any evidence. Author mentioned the homologous sequence of this transcript was not detected in other mammals, however, RNA structure of this lncRNA could be conserved in other mammals.

2, Authors did not provide negative control for their FISH experiments. To demonstrate the specificity of FISH probe, authors could include siRNA against NORFA and compare the signal and localization in the cells.

3, The center part of this manuscript is NORFA serve as a sponge for miR-126, although authors provide multiple line evidence, it is still not convincing. However, authors should perform RNA pull-down assay in the cells, at least by overexpressing NORFA and miR-126, rather relying on in vitro RNA binding assay.

More importantly, authors should mutate miR-126 binding sites in the construct of pcDNA3.1-NORFA and overexpress it in cells to see whether NORFA-mu could still be able to reduce the expression of miR-126 and other responding pathways.

4, Authors state this lncRNA has a important role for sow fertility, however, the evidence provided here is not sufficient to draw any conclusion on it.

Reviewers' comments:

Reviewer #1 (Remarks to the Author):

Du X et al. identified novel lincRNA, NORFA from SMAD4-silenced porcine GCs. They found that NORFA has anti-apoptotic activity in GC by functioning as a ceRNA of miR-126, which has pro-apoptotic activity in GC. They also found that the pro-apoptotic activity of miR-126 is via the direct targeting of TGFbeta-R2. They confirm that anti-apoptotic activity of NORFA is through the inhibition of miR-126 to target TGFbeta-R2. In addition, they found that NORFA promoter has 19-bp duplication that is targeted by transcriptional activator, NFIX, which is highly expressed in healthy follicles. Based on these results, they concluded that NORFA functions as an inhibitor of granulosa cell apoptosis.

Overall, the data is interesting and well designed to draw conclusion. However, I have 3-concerns that (1) some of subtitle is not matching with the data (2) The manuscript is too lengthy. I think that Figure 1-8 and 9-10 is different story. Is it necessary to combine these data all together in the manuscript? (3) Again, Discussion is too too lengthy. Discussion should be the discussion about the data in the manuscript.

Response: Thanks very much. According to your questions, we have revised our manuscript by (1) modifying several subtitles to make them match with the data, (2) explaining why Fig. 9-10 are necessary for this article, and (3) shortening the Discussion part. Besides, we have also answered the following questions point-by-point.

Detailed comments are as follow,

(1) line 87 and line 100: The subtitle and conclusion of this part needs to be edited. The data of figure 2 support that NORFA is essential for GC apoptosis but low expression of NORFA in follicular atresia (Fig. 1g, h) does not mean it is essential in follicular atresia.

Response: Thanks, we have edited the subtitle and conclusion parts in this paragraph in the revised manuscript according to your advice. The subtitle have changed to 'NORFA is involved in GC apoptosis and follicular atresia', and conclusion have

changed to 'All our data suggest that NORFA is essential for inhibiting GC apoptosis, and involved in follicular atresia of pigs.'

(2) line 96: What does it mean "NORFA is an anti-apoptotic epigenetic mediator" ?

Response: Based on the results of Fig. 2, we demonstrate that the normal expression of NORFA is necessary for inhibiting porcine GCs apoptosis, indicating that NORFA is an anti-apoptotic factor in porcine GCs. Besides, NORFA has been proved to be a non-coding RNA which mainly function as epigenetic factors.

In order to make it clear for readers, we have modified this sentence as 'indicating that NORFA is an anti-apoptotic factor in porcine GCs' in the revised manuscript.

(3) line 110: Please insert reference for "miR-126, an intronic miRNA transcript from EGFL7"

Response: Thanks. According to your advice, we have insert a reference for indicating this sentence 'miR-126, an intronic miRNA transcript from EGFL7'. The reference is listed below:

Zhang, Y. *et al.* miR-126 and miR-126* repress recruitment of mesenchymal stem cells and inflammatory monocytes to inhibit breast cancer metastasis. *Nat. Cell Biol.* **15**, 284-294 (2013).

(4) line 137: The subtitle "in vivo" is not appropriate.

Response: Thanks. In order to avoid misleading readers, we have replaced '*in vivo*' with 'in porcine ovarian follicles' in the revised manuscript.

(5) Line 159 and line 176: The subtitle and conclusion of figure-5 is confusing. miR-126 induces GC apoptosis and NORFA inhibits pro-apoptotic activity of miR-126, right?

Response: Thanks very much, we have revised the subtitle and conclusion of Fig. 5 according to your advice in the revised manuscript.

(6) Line 199: Figure 6g and 6i, Western band seems to be saturated. Can we expect

more clear western result by reducing the amount of total protein in the western blot analysis?

Response: Thanks. In order to get more clear bands, we have re-performed the western blotting assays by reducing the amount of total protein according to your advice. As shown in Fig. R1 as well as Fig. 6 in the revised manuscript, we demonstrate that ectopic expression of miR-126 dramatically reduces TGFBR2 protein level in porcine GCs, while the opposite result was observed after miR-126 silencing.

Fig. R1. miR-126 inhibits TGFBR2 expression in porcine GCs. (a) The protein level of TGFBR2 in porcine GCs treated with miR-126 mimics was measured by western blotting assay. (b) TGFBR2 protein level in porcine GCs after transfection with miR-126 inhibitor was detected by western blotting assay.

(7) Line 229 and Line 253: The subtitle and conclusion of figure-8 is confusing. miR-126 inhibits the NORFA activity to enhance TGF- β signaling, right?

Response: Thanks. We have modified the subtitle and conclusion parts of Fig. 8 in the revised manuscript in order to make the meaning clear for readers. The new version of subtitle is 'NORFA activates TGF- β signaling pathway by miR-126-TGFBR2 axis', and the new version of conclusion is 'miR-126-TGFBR2 axis is involved in the regulation of NORFA to TGF- β signaling pathway in porcine GCs.'

(8) Fig 8-H, I, K: The difference of band intensity between control and test is marginal.

Can we expect more clear western result by reducing the amount of total protein in the western blot analysis?

Response: Thanks. According to your advice, we have re-performed the western blotting assays in Fig. 8h, i, k by reducing the amount of total protein in order to reflect the difference of band intensity. The results are represented in Fig. R2 here as well as Fig. 8 in the revised manuscript.

Fig. R2. Reconfirm the difference of band intensity between control and test in Fig. 8h, i, k. (a-c) Western blotting assays were re-performed under the same condition with Fig. 8h, i and k except for lower amount of total loading protein.

(9) Figure-9 and -10: I think that the conclusion of the manuscript is solid with figure 1-8.

Response: Thanks. First, according to the results of Fig.1-8, we identified a pig-specific novel lncRNA, NORFA, and demonstrated that it inhibits porcine GCs apoptosis and is involved in follicular atresia by activating TGF- β signaling pathway through interacting with miR-126-TGFBR2 axis. This part mainly focus on investigating the functions and mechanisms of NORFA in modulating GC apoptosis. As we know, follicular atresia is a limiting factor for female infertility. Thus, we hypothesized that NORFA might participate in regulating sow fertility and begin to investigate the relationship between NORFA and sow fertility, and the results are shown in Fig. 9-10. Based on Fig. 9-10, we identified a 19-bp breed-specific duplication in the core promoter of NORFA could affect its transcription, and found that it is involved in sow

fertility. This part (Fig. 9-10) mainly focuses on investigating the important role of *NORFA* in sow breeding works. The combination of two parts makes this article more comprehensive and contributes to a depth understanding of the functions of *NORFA* and its role in regulating female reproduction traits. For these reasons, we believe that Fig. 9-10 is especially important and indispensable for this article.

(10) Discussion is too lengthy.

Response: Thanks. According to your advice, we have shorten the discussion part in the revised manuscript as following:

Delete "thereby improving the 100,000 in cattle and sheep." in Line 319-321.

Delete "miR-1306³³, and miR-144³⁴." in Line 325.

Delete "Besides, lnc-mg has been and miR-351-5p^{44, 45}." in Line 350-352.

Delete "especially in cancer cells,.....Previous studies reported that" in Line 370-375.

Delete "For instance,by TGF- β 1 in porcine GCs^{29, 33}" in Line 378-380.

Delete "Besides, recent study..... by hosting miR-675⁵⁹" in Line 388-390.

Delete ", two variants rs34552516..... and type 1 diabetes⁶²" in Line 403-404.

Delete "LncOb, a fat-specific of leptin and obesity⁶⁴" in Line 405-407.

Delete ", such as cerebellum Malan syndrome⁷⁵ and cancer⁷⁶" in Line 429-431.

Reviewer #2 (Remarks to the Author):

Comment and suggestions for authors:

The manuscript “NORFA, a novel candidate lincRNA for sow fertility, inhibits granulosa cell apoptosis” describes the role of a lincRNA, NORFA in porcine granulosa cell (GC) apoptosis, follicular atresia, and sow fertility. The study sort to examine if NORFA was involved in GC apoptosis and follicular atresia and to determine the actual role it plays and its mechanism of action. The study results and discussion propose that NORFA sponges endogenous miR-126 in porcine GCs and prevent its binding to the 3' UTR of TGFBR2, releasing TGFBR2 to inhibit GC apoptosis and follicular atresia. The study also identified a 19-bp duplication in the promoter region of NORFA which is a sow prolificacy-associated variant that recruits the transcription factor NF1X to enhance NORFA transcription and regulation of GC apoptosis and follicular atresia.

The study was thoroughly conducted with adequate samples, replicates, controls. The aim is clear and technically sound methodology was used to arrive at the conclusion. Sufficient data have been provided to support the claims of the study and the data is made available. The discussion is elaborate and the conclusion is drawn from the purpose of the study and the results obtained from the experiments. Appropriate references have been cited when necessary in most cases for a comprehensive understanding of the study.

The manuscript is written in standard English.

However, the following revisions need to be made.

Response: Thanks very much. According to your suggestions, we have revised our manuscript and answered the following questions point-by-point.

Specific comments

Revise the following statements at abstract to make them meaningful and clear;

- The first sentence in the abstract (line 8) does not express any coherent idea. It seems to suggest that “lincRNAs have been implicated in healthy and disease

conditions” . Restructure the statement to reflect so, if that is what the sentence seeks to suggest.

Response: Thanks. To make it clear for readers, we have restructured the statement as ‘lincRNAs have been proved to be involved in regulating health and disease in organisms’ in the revised manuscript according to your advice.

- Line 15-16 (furthermore, the correlations among NORFA, miR-126 and TGFBR2 levels were validated in follicles. SUGGESTION: The correlation between NORFA, miR-126, and TGFBR2 levels in follicles was further validated)

Response: Thanks. We have revised this sentence according to your suggestion.

The introduction or background of the study has no heading/subheading (line 22).

Response: Thanks. We have added the heading ‘**Introduction**’ (line 22) before the introduction part in the revised manuscript according to your suggestion.

Cite reference for the ideas expressed in the statement at lines 35 to 37 of introduction.

Response: Thanks. We have cited reference for this sentence in the revised manuscript according to your suggestion. The cited reference was shown below:

Gebert, L.F.R. & MacRae, I.J. Regulation of microRNA function in animals. *Nat Rev Mol Cell Biol* **20**, 21-37 (2019).

At results section, line 108-109, the statement is ambiguous, restructure it to indicate the expression level of the four genes were increased.

Response: Thanks. To make it clear for readers, we have revised it as ‘The expression levels of four coding genes (*EGFL7*, *PHPT1*, *TMEM141* and *LCN10*) were increased in GCs after *NORFA* overexpression (Fig. 3b), but decreased after *NORFA* silencing (Fig. 3c).’ according to your advice.

The meaning of the statement at discussion (line 349-350) is not clear. Revise it to

capture what you want to you want to express.

Response: Thanks. To make it clear for readers, we have replaced this sentence as 'Acting as a ceRNA is the main function mode for lincRNAs containing the same miRNA response elements (MREs) with targets' in the revised manuscript.

Minor suggestions

Below are some recommended suggestions for some words or statements in the manuscript:

Replace “we report a novel lincRNA” (line10) with (we report that a novel lincRNA),” . Replace “Prevented” with “preventing” (line 13). “Are identified” with “have been identified” (line29-30). Put “are” between miRNAs and mainly (line 35). Replace “which” with “with” and “show” with “showing” (line 38). “To” with “and” (line 49). “It is little known about” should be replaced with “little is known of” (line 51).

Response: Thanks. We have revised these statements according to your suggestion.

At results;

Change “with highly expressed” to “which was highly expressed” (line 64). Replace “is” with “to be” and “to locate” with “to be located” (line 67). Put “the” before “genome” (line 70). Replace “the” with “a’ (line 71). Put “more” before “especially (line 75). Put a comma after “that” (line 90). Put “our” after “all” (line 100). “comprising” is much more appropriate than “including” (line 107). Put a semicolon after “including” (line 108). Change “were decreased” to “showed decreased expression” (line 109-110). Put “the” after “and” (line 113).

Put a full stop after “GCs” and begin the next sentence with a capital (line115). Put “of the” before NORFA (line 121). Change “physically” to “physical” (line 127). Replace “not” with “no” (line131).

Rephrase this statement at lines 137-138; “due to the lack of the characterization of

the gene encoding miR-126 in pig” . SUGGESTION: change “the lack of” to “the unavailability of data or information on” . Change “highly” to “high” (line 140). Put “is” before “consistent” (line 151).

Change “we next to analyse” to “next we analyzed” (line 159-160). Replace “positive” with “positively” (line 232). Do same on line 233. Change “detected” to “examined the” (line 242). Replace “which” with “was” (line 252). Change “breeds” to “breed” (line 262). Put “of” in front of “which” (line 268). Change “only exists” to “exists only” (line 268). Change “investigate” to “investigation” (line 281).

Response: Thanks. We have revised these statements in the result part according to your suggestion.

At discussion;

Put “of” in front of “follicles” (line 318). Put “the” in front of “mechanism” (line 319). Replace “reproductive” with “reproduction” (line 333). Change “that the” to “whose” (line 334), “opposite” to “oppositely” (line 356). Put a semicolon after “including” (lines 362 and 364). Change “we showed” to “we have shown” (line 367). Replace “was” with “has been” (line 384). Change “have” to “have been” (line 402). Change “a biomarker” to “biomarkers” , and “disease” to “diseases” (line 403). Change “identify” to “identified” (line 415). Change “functions” to “function” (line 432), and “direct” to “directly” (line 433). Remove “the” (line 436). Put “is” after “which” (line 452).

Response: Thanks. We have revised these statements in the discussion part according to your suggestion.

At method;

Remove “through” (line 510). Change “end” to “ends” (line 524). Put “was” before the semicolons on lines 529 and 530. Replace “as” with “by” (line 557).

Change “presenting” to “present” (line 615). Replace “served” with “use” (line 620). Put “tissue” in front of “sample” (line 632).

Response: Thanks. We have revised these sentence in the method part according to your suggestion.

Supplementary data

At supplementary figure 6, “schematic” should be “schematic diagram” .

Response: Thanks. We have added ‘diagram’ after ‘schematic’ in the figure legend of supplementary figure 6 according to your suggestion.

Reviewer #3 (Remarks to the Author):

In this manuscript by Du et al., authors identified that pig-specific lncRNA NORFA can regulate granulosa cell apoptosis by acting as a sponge for miR-126, and further demonstrated that NORFA/miR-126 axis plays an important role in regulating GCs apoptosis through targeting TGFBR2. In the end, authors identified a pig-specific 19-bp duplication in NORFA promoter, which could regulate NORFA transcription by altering the recruitment of NFIX to the promoter of NORFA.

Overall, this study is very interesting and authors provided comprehensive experiments. However, the results were not solid enough to support the conclusions. Below are my major comments.

1, Authors stated this lncRNA is pig-specific, but did not provide any evidence. Author mentioned the homologous sequence of this transcript was not detected in other mammals, however, RNA structure of this lncRNA could be conserved in other mammals.

Response: Thanks. According to your advice, we have analyzed the structure conservation of porcine NORFA among other mammal species with the strategy below (Fig. R3a). First, we analyzed the secondary structure of porcine NORFA using two software (RNAfold and RNAstructuer including SHAPE-map functions). As shown in Fig.R3b, 29 helices and 4 junctions were identified within porcine NORFA but none

of which is high conservative among other species (H21 for example, Fig.R3c). Second, the tertiary structure and the domains of NORFA were predicted using RNAcomposer. Three domains were predicted (D1: 44nt-192nt, D2: 224nt-515nt, D3: 546nt-661nt) and we noticed that three domains just had low conservation among different mammal species (Fig.R3d). In addition, we also analyzed the similarity of the potential open reading frame (ORF, only one ORF consisted by 168 nt was identified in the reversed strand) of porcine *NORFA* but its conservative was low (<25%) among other species (Fig.R3e). All the data above demonstrate that porcine *NORFA* only has low structure conservation.

On the other hand, It is worth noting the primary structure (sequence) of genes determine their secondary, tertiary and even space structure. Furthermore, the similarity of gene structure determined the conservation of their domains and functions among different species. To our knowledge, the conservation of lncRNA is usually determined by their primary structure. Besides, we also noticed that in order to analyze the conservation of lncRNAs, it is necessary to analyze their sequence and chromosome locations¹⁻⁵. Thus, the primary conservation and chromosome location of *NORFA* were analyzed and we found that the primary structure of *NORFA* has low similarity and the desert region between *EDF1* and *TRAF2* (two neighbour genes around porcine *NORFA*) among different species are not conserved (Fig.R3f, g). According to the findings, we draw the conclusion that *NORFA* is pig-specific lncRNA.

c

Human (hg38) BLAT Results	Mouse (mm10) BLAT Results																								
Sorry, no matches found (scoring higher than 20)	Sorry, no matches found (scoring higher than 20)																								
Chimp (panTro6) BLAT Results	Sheep (oviAri4) BLAT Results																								
Sorry, no matches found (scoring higher than 20)	Sorry, no matches found (scoring higher than 20)																								
Cow (bosTau9) BLAT Results	Chicken (galGal6) BLAT Results																								
Sorry, no matches found (scoring higher than 20)	Sorry, no matches found (scoring higher than 20)																								
Fig (susScr1) BLAT Results																									
   ACTIONS QUERY SCORE START END QSIZE IDENTITY CBROW STRAND START END SPAN     browser details YourSeq 36 1 36 36 100.0% chr16_NF_01808483v1 - 947947 947982 36   	ACTIONS	QUERY	SCORE	START	END	QSIZE	IDENTITY	CBROW	STRAND	START	END	SPAN	browser details	YourSeq	36	1	36	36	100.0%	chr16_NF_01808483v1	-	947947	947982	36	
ACTIONS	QUERY	SCORE	START	END	QSIZE	IDENTITY	CBROW	STRAND	START	END	SPAN														
browser details	YourSeq	36	1	36	36	100.0%	chr16_NF_01808483v1	-	947947	947982	36														

D1 domain

Human (hg38) BLAT Results

BLAT Search Results

ACTIONS	QUERY	SCORE	START	END	QSIZE	IDENTITY	CBROW	STRAND	START	END	SPAN
browser details	NORFA	20	95	117	148	100.0%	chr17	+	10198866	10198885	20

D2 domain

Human (hg38) BLAT Results

BLAT Search Results

ACTIONS	QUERY	SCORE	START	END	QSIZE	IDENTITY	CBROW	STRAND	START	END	SPAN
browser details	NORFA	33	124	290	292	62.2%	chr2	+	206030434	206030541	108
browser details	NORFA	22	121	143	292	100.0%	chr20	+	51270275	51270301	27
browser details	NORFA	20	122	141	292	100.0%	chr4	-	168879999	168880018	20

D3 domain

Human (hg38) BLAT Results

BLAT Search Results

ACTIONS	QUERY	SCORE	START	END	QSIZE	IDENTITY	CBROW	STRAND	START	END	SPAN
browser details	NORFA	29	19	56	116	94.2%	chr3	+	181696143	181696188	46

e

ORF Finder results

Results for 739 residue sequence "NORFA" starting "CATCAGCCTC"

No ORFs were found in reading frame 1.

ORF Finder results

Results for 739 residue sequence "NORFA" starting "CATCAGCCTC"

NORF number 1 in reading frame 1 on the reverse strand extends from base 92 to base 249
 ATGAGGTTTCTCTTAACCCACAGCTAAGCCAGCTTAACAGCTAAGAGATCTCTCT
 CCGGAGCCCTGGAGGCGCGGCGCCCTCTGACCGCTCCACAGGTTCTCTCCAGAG
 GGAGTTCACCTCCAGCAGCTGACCCCTCTTCTTCCAGGTAATGA

>Translation of ORF number 1 in reading frame 1 on the reverse strand.
 MFLSNFTMFAVLTLKGSFPGAGPQPSCDASAGFLIPLSAGTAAALRFPPFVWV

All Genomes BLAT Results

Name	Genome	Assembly	Tiles	Chrom
NORFA Pig		susScr11	31	chrUn_NW_018084833v1
NORFA American alligator		allMis1	4	JH734654
NORFA Bison		bisBis1	4	KN264896v1
NORFA Cow		bosTau9	4	chr16
NORFA Horse		equCab3	4	chr24
NORFA Chicken		galGal6	4	chr19
...		...		
NORFA Gorilla		gorGor5	2	CYUI01000003v1
NORFA Human		hg38	2	chr1
NORFA Coelacanth		latCha1	2	JH126564
NORFA Elephant		loxAfr3	2	scaffold_0

Cow (bosTau9) BLAT Results

BLAT Search Results

ACTIONS	QUERY	SCORE	START	END	QSIZE	IDENTITY	CBROW	STRAND	START	END	SPAN
browser details	NORFA	26	141	166	168	100.0%	chr11	-	105959425	105959450	26
browser details	NORFA	23	129	154	168	96.2%	chr11	+	78399566	78399593	28
browser details	NORFA	20	109	128	168	100.0%	chr24	+	42274447	42274466	20

Chicken (galGal6) BLAT Results

BLAT Search Results

ACTIONS	QUERY	SCORE	START	END	QSIZE	IDENTITY	CBROW	STRAND	START	END	SPAN
browser details	NORFA	24	120	144	168	100.0%	chr3	+	57619879	57619906	28
browser details	NORFA	23	119	146	168	96.0%	chr14	+	7090096	7090124	29
browser details	NORFA	23	119	146	168	96.0%	chr14	+	7186580	7186608	29
browser details	NORFA	22	120	143	168	95.9%	chr2	+	8027210	8027233	24
browser details	NORFA	20	66	85	168	100.0%	chr25	-	1429916	1429935	20
browser details	NORFA	20	127	146	168	100.0%	chr1	-	663655	663674	20

Human (hg38) BLAT Results

BLAT Search Results

ACTIONS	QUERY	SCORE	START	END	QSIZE	IDENTITY	CBROW	STRAND	START	END	SPAN
browser details	NORFA	29	58	95	168	94.2%	chr3	-	181696143	181696188	46
browser details	NORFA	23	124	147	168	100.0%	chr17	+	73451838	73451863	26
browser details	NORFA	20	138	157	168	100.0%	chr8	-	18171443	18171462	20

f

All Genomes BLAT Results

Name	Genome	Assembly	Tiles	Chrom
NORFA	Pig	susScr11	144	chrUn_NW_018084833v1
NORFA	Cow	bosTau9	8	chr11
NORFA	Sheep	oviAri4	6	chr3
NORFA	Chicken	galGal6	4	chr1
NORFA	Human	hg38	4	chr2
NORFA	Mouse	mm10	4	chr16

Cow (bosTau9) BLAT Results

BLAT Search Results

ACTIONS	QUERY	SCORE	START	END	QSIZE	IDENTITY	CHROM	STRAND	START	END	SPAN
browser details	NORFA	56	50	718	739	49.5%	chr11	+	105958875	105959819	945
browser details	NORFA	22	301	323	739	100.0%	chr21	-	67144106	67144134	29
browser details	NORFA	21	313	333	739	100.0%	chr17	-	67110189	67110209	21
browser details	NORFA	20	576	595	739	100.0%	chr21	+	69527059	69527078	20
browser details	NORFA	20	164	183	739	100.0%	chr13	+	62677480	62677499	20
browser details	NORFA	20	6	25	739	100.0%	chr12	+	35225350	35225369	20

Sheep (oviAri4) BLAT Results

BLAT Search Results

ACTIONS	QUERY	SCORE	START	END	QSIZE	IDENTITY	CHROM	STRAND	START	END	SPAN
browser details	NORFA	92	50	525	739	47.0%	chr3	-	812354	812897	544
browser details	NORFA	24	1	25	739	100.0%	chr19	-	60265655	60266028	374
browser details	NORFA	24	434	464	739	44.1%	chr18	+	56438471	56438497	27
browser details	NORFA	23	189	216	739	88.0%	chr14	-	14071843	14071869	27
browser details	NORFA	22	441	466	739	92.4%	chr20	+	14229295	14229320	26
browser details	NORFA	22	2	24	739	100.0%	chr2	+	229784821	229784846	26
browser details	NORFA	20	191	210	739	100.0%	chr20	-	48691001	48691020	20
browser details	NORFA	20	164	183	739	100.0%	chr13	+	62004179	62004198	20

Human (hg38) BLAT Results

BLAT Search Results

ACTIONS	QUERY	SCORE	START	END	QSIZE	IDENTITY	CHROM	STRAND	START	END	SPAN
browser details	NORFA	27	31	58	739	100.0%	chr17	+	33333051	33333088	38
browser details	NORFA	24	486	510	739	100.0%	chr15	-	47689554	47689596	43
browser details	NORFA	23	512	535	739	100.0%	chr17	-	73451838	73451863	26
browser details	NORFA	20	141	160	739	100.0%	chr17	+	10198866	10198885	20
browser details	NORFA	20	35	56	739	95.5%	chr11	+	40930981	40931002	22

Mouse (mm10) BLAT Results

BLAT Search Results

ACTIONS	QUERY	SCORE	START	END	QSIZE	IDENTITY	CHROM	STRAND	START	END	SPAN
browser details	NORFA	28	499	539	739	93.8%	chr14	+	11448625	11448672	48
browser details	NORFA	25	504	539	739	81.5%	chr17	+	46612779	46612811	33
browser details	NORFA	25	633	658	739	100.0%	chr10	+	60738891	60738922	32
browser details	NORFA	24	310	334	739	100.0%	chr13	+	69969870	69969913	44
browser details	NORFA	23	359	382	739	100.0%	chr12	-	71995296	71995331	36
browser details	NORFA	21	223	243	739	100.0%	chr6	+	64452496	64452516	21
browser details	NORFA	21	47	69	739	95.7%	chr13	+	17404261	17404283	23

g

Fig. R3. Identification of the conservation of pig NORFA structure. (a) Diagram depicting the investigation strategy for pig NORFA conservation. (b) The secondary structure of pig NORFA was analyzed by RNAfold and RNAstructure software. H: helice, J: junction. (c) The conservation of NORFA secondary structure, diagram showing the conservation of H21 of pig NORFA by UCSC database among different species. (d) The tertiary structure and domains of pig NORFA were predicted and the conservation of these domains were detected using UCSC. (e) The potential ORFs within pig NORFA (sense and anti-sense) were predicted and their conservation were detected by UCSC. (f) The conservation of pig NORFA primary structure was detected. (g) The conservation of desert region between *EDF1* and *TRAF2* (red box) among different mammal species were detected.

Reference:

1. Lin, Y., Schmidt, B.F., Bruchez, M.P. & McManus, C.J. Structural analyses of NEAT1 lncRNAs

suggest long-range RNA interactions that may contribute to paraspeckle architecture. *Nucleic Acids Res* **46**, 3742-3752 (2018).

2. Owens, M.C., Clark, S.C., Yankey, A. & Somarowthu, S. Identifying Structural Domains and Conserved Regions in the Long Non-Coding RNA IncTCF7. *Int J Mol Sci* **20** (2019).
3. Sherpa, C., Rausch, J.W. & Le Grice, S.F. Structural characterization of maternally expressed gene 3 RNA reveals conserved motifs and potential sites of interaction with polycomb repressive complex 2. *Nucleic Acids Res* **46**, 10432-10447 (2018).
4. Tunnicliffe, R.B., Levy, C., Ruiz Nivia, H.D., Sandri-Goldin, R.M. & Golovanov, A.P. Structural identification of conserved RNA binding sites in herpesvirus ORF57 homologs: implications for PAN RNA recognition. *Nucleic Acids Res* **47**, 1987-2001 (2019).
5. Uroda, T. *et al.* Conserved Pseudoknots in lncRNA MEG3 Are Essential for Stimulation of the p53 Pathway. *Mol Cell* **75**, 982-995 e989 (2019).

2, Authors did not provide negative control for their FISH experiments. To demonstrate the specificity of FISH probe, authors could include siRNA against NORFA and compare the signal and localization in the cells.

Response: Thanks. According to your advice, we have complemented the negative control for our FISH experiments, see more details in Fig. R4a. Besides, we have also performed FISH in porcine GCs after *NORFA* silencing. As shown in Fig. R4b, the signals of *NORFA* in the porcine GCs were dramatically reduced (24h, 36 h and 48 h) and we also noticed that the reduced signals mainly exist in the cytoplasm of porcine GCs, indicating that the FISH probe has high specificity and also identified that *NORFA* is mainly located at cytoplasm of porcine GCs.

Fig. R4. Detection the specificity of *NORFA* FISH probe. (a) The negative control for RNA FISH assays. (b) The signals of *NORFA* in porcine GCs after transfection with *NORFA*-siRNA (si*NORFA*) for different time (24 h, 36 h and 48 h) were detected by FISH assays.

3, The center part of this manuscript is *NORFA* serve as a sponge for miR-126, although authors provide multiple line evidence, it is still not convincing. However, authors should perform RNA pull-down assay in the cells, at least by overexpressing *NORFA* and miR-126, rather relying on in vitro RNA binding assay.

More importantly, authors should mutate miR-126 binding sites in the construct of pcDNA3.1-*NORFA* and overexpress it in cells to see whether *NORFA*-mu could still be able to reduce the expression of miR-126 and other responding pathways.

Response: Thanks. According to your advice, we have performed the RNA pull-down assay *NORFA* or miR-126 overexpressed porcine GCs. As shown in Fig. R5a, b, the enrichment of miR-126 in *NORFA* overexpressed porcine GCs is reduced, but increased in miR-126 treated porcine GCs in comparison with control group.

Besides, we have constructed the pcDNA3.1-*NORFA*-mut vector containing mutant type miR-126 response element (Fig. R5c) and transfected it into porcine GCs. As shown in Fig. R5d, e, overexpression of mutated *NORFA* has no effect on the

expression of miR-126 and TGFBR2 at RNA level, and also the TGFBR2 and p-SMAD3 protein levels.

Fig. R5. NORFA interacts with miR-126 in porcine GCs. (a) Left panel: the expression levels of *NORFA* and miR-126 in *NORFA* overexpressed porcine GCs were detected by qRT-PCR. Right panel: the enrichment of miR-126 on biotin-labeled *NORFA* in *NORFA* overexpressed porcine GCs was detected by RNA pull-down. (b) Left panel: the expression levels of miR-126 and *NORFA* in miR-126 overexpressed porcine GCs were detected by qRT-PCR. Right panel: the enrichment of miR-126 on biotin-labeled *NORFA* in miR-126 overexpressed porcine GCs was detected by RNA pull-down. (c) Construction of pcDNA3.1-NORFA-mut vector with miR-126 response element mutation. (d) The expression levels of *NORFA*, miR-126 and *TGFBR2* in porcine GCs transfected with pcDNA3.1-NORFA-mut vectors were measured by qRT-PCR. (e) The protein levels of TGFBR2 and p-SMAD3 in porcine GCs treated with NORFA^{OE} or NORFA-mut^{OE} were detected by western blot. Data were shown as mean ± SEM with three independent experiments. *P* values were calculated by using a two-tailed Student's *t*-test. **P*<0.05, ***P*<0.01 and ns indicates no significance.

4, Authors state this lncRNA has a important role for sow fertility, however, the evidence provided here is not sufficient to draw any conclusion on it.

Response: Thanks. In this study, we have proved that *NORFA* suppressed porcine GC

apoptosis through miR-126-TGFBR2-SMAD3 pathway signaling axis by using gain-or-loss functions. To our knowledge, GC apoptosis is the main cause of follicular atresia in mammal female ovaries¹⁻³. Furthermore, we have also demonstrated that NORFA is differentially expressed in healthy (with high level of NORFA) and atretic (with relative low level of NORFA) follicles, suggesting that NORFA is involved in sow follicular atresia by inhibiting GC apoptosis. It has been reported that high-prolific pig breeds represent low atretic follicles, while low-prolific breeds show relative high atretic follicles⁴. Together, these results functionally prove that NORFA is an anti-apoptotic factor in porcine GCs which further relate to sow follicular atresia and fertility.

In addition, we also found that the expression level of NORFA in Erhualian (a Chinese famous pig breed with the highest born number record) follicles is higher than that in Large White (a European pig breed with relative low prolific performance) follicles at all stages during follicular development. Furthermore, we have also proved that the 19-bp duplication mutation in the promoter of *NORFA* leads to its high expression level by recruiting more NFIX, which functions as a transcription factor. Overall, these findings genetically demonstrate that NORFA is a candidate factor closely association with sow fertility.

Reference:

1. Matsuda, F., Inoue, N., Manabe, N. & Ohkura, S. Follicular growth and atresia in mammalian ovaries: regulation by survival and death of granulosa cells. *J Reprod Dev* **58**, 44-50 (2012).
2. Asselin, E., Xiao, C.W., Wang, Y.F. & Tsang, B.K. Mammalian follicular development and atresia: role of apoptosis. *Biol Signals Recept* **9**, 87-95 (2000).
3. Jiang, J.Y., Cheung, C.K., Wang, Y. & Tsang, B.K. Regulation of cell death and cell survival gene expression during ovarian follicular development and atresia. *Front Biosci* **8**, d222-237 (2003).
4. Miller, A.T., Picton, H.M., Craigon, J. & Hunter, M.G. Follicle dynamics and aromatase activity in high-ovulating Meishan sows and in Large-White hybrid contemporaries. *Biol Reprod* **58**, 1372-1378 (1998).

REVIEWERS' COMMENTS:

Reviewer #1 (Remarks to the Author):

Reviewer's comments are well addressed in the revised manuscript and the manuscript is significantly improved by the revision.

Reviewer #3 (Remarks to the Author):

Authors have addressed all my concerns.

However, authors should include all the revised results into either main figures or supplemental figures, and revise the figures and main text accordingly.

Presenting the revised results only in the rebuttal letter will not be accessible for general readers.

I will recommend for publication after authors revising this.

Rebuttal Letter

Reviewer #1 (Remarks to the Author):

Reviewer's comments are well addressed in the revised manuscript and the manuscript is significantly improved by the revision.

Response: Thanks very much for your valuable comments.

Reviewer #3 (Remarks to the Author):

Authors have addressed all my concerns.

However, authors should include all the revised results into either main figures or supplemental figures, and revise the figures and main text accordingly. Presenting the revised results only in the rebuttal letter will not be accessible for general readers.

I will recommend for publication after authors revising this.

Response: Thanks very much, we have incorporated the revised results into the main text and supplementary information in the final version of our text according to your advice.